# Understanding Federated Unlearning through the Lens of Memorization

## Abstract

Federated learning (FL) must support unlearning to meet privacy regulations. However, existing federated unlearning approaches may overlook the overlapping information between the unlearning and remaining data, leading to ineffective unlearning and unfairness between clients. We revisit this problem through the lens of memorization, showing that only unique memorization information from the unlearning dataset should be removed, while shared patterns should remain. Subsequently, we propose the Grouped Memorization Evaluation, a metric that distinguishes memorized from shared knowledge at example level, and introduce Federated Memorization Eraser (FedMemEraser), a pruning-based method that resets redundant parameters carrying memorization information. The experimental results demonstrate that our method closely matches the retraining baselines and effectively eliminates memorization information compared to other unlearning algorithms.

## 1 Introduction

Federated learning (FL) has become a popular machine learning paradigm in recent years (McMahan et al., 2017). It enables collaborative model training without sharing raw data, where participants train locally and exchange only model updates. An essential requirement of federated learning is federated unlearning (FU) (Kairouz et al., 2021). This concept is referred to as the right to be forgotten (RTBF) (Liu et al., 2022), as mandated by privacy regulations such as the General Data Protection Regulation (GDPR) (Voigt & Von dem Bussche, 2017) and the California Consumer Privacy Act (CCPA) (Harding et al., 2019). Consequently, incorporating unlearning mechanisms into FL is essential to maintain user privacy and meet legal requirements.

A straightforward approach for federated unlearning is to retrain the federated learning model. However, retraining involves significant computational and communication costs. Consequently, performing unlearning directly on the original model is a more efficient way. Several federated unlearning (FU) techniques have been proposed to address this challenge. For example, perturbing information representations can facilitate unlearning. Gu et al. (2024) fine-tune the original model using a randomly labeled unlearning dataset to compromise the learned representations. Furthermore, historical information can also support unlearning. FedRecovery (Zhang et al., 2023) employs differential privacy and historical updates to make the unlearning data indistinguishable. Additionally, gradient ascent is another widely adopted strategy. Halimi et al. (2022) apply this technique to reverse the learning process, incorporating a $l_2$ norm constraint to prevent arbitrary updates. FedOSD (Pan et al., 2025) modifies the loss function and addresses gradient conflicts during unlearning to retain generalization performance. Besides, the loss function optimization based on the Fisher Information Matrix (FIM) (Liu et al., 2022) has proven effective in guiding unlearning.

However, existing federated unlearning algorithms may overlook the overlapping learnable information between the unlearning and remaining clients. For example, Pan et al. (2025) and Halimi et al. (2022) attempt to eliminate the influence of the entire unlearning dataset, including the overlapping information. Such removal or oversight can result in ineffective unlearning and unfairness among clients. As illustrated in Figure 1, under a CIFAR-10 Non-IID setup with class 1 data, the embedding features of the unlearning client (light red) still exhibit substantial overlap with those of the remaining clients (light green and blue) in the retrained model. Moreover, the overlapping data points are correctly classified and not forgotten, even if the data come from the unlearning

client. This indicates that overlapping data encode general, shared knowledge derived from both the unlearning client and the remaining clients. Therefore, the overlapping portion may substantially contribute to the model's generalization performance, and its removal could potentially degrade the overall generalization capability of the model. Furthermore, non-overlapping information from the unlearning dataset may be overlooked and inadvertently retained in the unlearned model in unlearning. On the other hand, unlearning overlapping information may undermine the contributions of clients that provide similar information, while clients contributing distinct information remain unaffected. For example, as illustrated in Figure 1, removing overlapping features may lead to a more significant performance degradation for remaining client 1 (light blue) compared to remaining client 0 (light green), since client 1 provides more overlapping features. This can result in inconsistent performance shifts across the remaining clients and introduce unfairness.

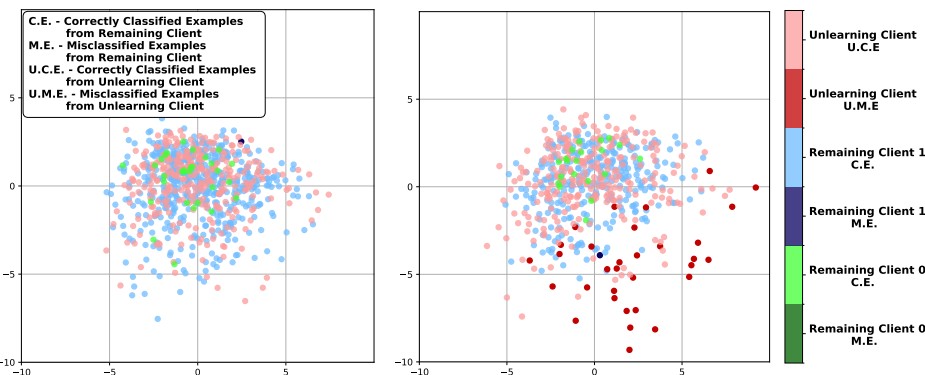

Figure 1: Embedding feature distribution (only class 1) of clients on original model (Left) and retrained model (Right).

To address these problems, we revisit the federated unlearning problem. From a memorization perspective, the overlapping features represent shared and generalized information found in both the unlearning and retained datasets. In contrast, the non-overlapping features are unique and memorized information provided in corresponding datasets. Based on this insight, we find the key difference at the information level between the original model and the retrained model lies in the memorization information contained within the unlearning dataset. Therefore, we propose federated memorization unlearning, which demonstrates that **only the unique memorization specific to the unlearning dataset needs to be removed, rather than the entire unlearning dataset**. Given the lack of a reliable memorization metric in unlearning, we propose a novel evaluation metric based on the memorization score (Feldman & Zhang, 2020), enabling finer-grained assessment of unlearning effectiveness. In addition, we present a novel federated unlearning method which only removes the memorization information: FedMemEraser. We identify redundant parameters as the primary carriers of memorization information, making them key targets for removal in unlearning. As a result, we retain the overlapping information in the unlearned model to preserve generalization performance and ensure fairness among clients. In the experiment, we demonstrate that our approach closely matches the retrained baselines and outperforms other baselines in unlearning efficacy, generalization performance, and client fairness.

Specifically, our contributions can be summarized as follows:

- We redefine the federated unlearning problem from the perspective of memorization and demonstrate that overlapping or shared information, should not be unlearned.

- We propose **Grouped Memorization Evaluation**, a novel metric that can measure memorization information at example level, thereby enabling a fine-grained assessment of unlearning efficacy.

- We introduce the **Federated Memorization Eraser**, a method designed to selectively only remove memorization information during unlearning. This approach preserves shared, overlapping knowledge, thereby maintaining generalization performance and promoting fairness across clients.

## 2 RELATED WORKS

### 2.1 MACHINE UNLEARNING

Machine unlearning (MU) aims to selectively remove the influence of specific training samples from a trained model, making it behave as if those samples were never included (Cao & Yang, 2015). While retraining from scratch guarantees exact removal (Wang et al., 2024), it is prohibitively expensive for large deep models, motivating more efficient MU methods. According to the taxonomy of machine unlearning (Li et al., 2025), approaches fall into exact and approximate unlearning. A notable exact method is SISA (Bourtoule et al., 2021), which partitions data into shards and retrains only the affected sub-model upon deletion. Approximate unlearning includes gradient-based methods such as negative gradient descent (Jang et al., 2022), random label perturbation (Fan et al., 2024), and influence function-based approaches (Guo et al., 2019; Golatkar et al., 2020), which leverage the Hessian or Fisher information to revert models toward an unlearned state.

### 2.2 MEMORIZATION EFFECT

Recent studies (Zhang et al., 2021; Feldman, 2020; Feldman & Zhang, 2020) reveal that DNNs often memorize specific details instead of learning general patterns, impacting generalization, security, and privacy, and closely relating to unlearning. Feldman (2020); Feldman & Zhang (2020) propose the memorization score to measure memorization and find the long-tail theory: DNNs tend to memorize atypical examples, which aid generalization but also increase privacy risks. Empirical evidence (Carlini et al., 2023; 2019) shows that memorization drives privacy vulnerabilities. Maini et al. (2023) locate memorization within a small subset of neurons across layers and show that frequent parameter updates mitigate it.

### 2.3 FEDERATED UNLEARNING

Federated unlearning (FU) extends machine unlearning to federated learning settings. Following a recent survey (Jeong et al., 2024), existing FU methods can be grouped into 4 main categories. The first category is based on historical information, aiming to revert the model to a pre-learning state. A typical approach is SISA (Bourtoule et al., 2021), which clusters clients and removes clusters containing the target clients during the unlearning process. Furthermore, historical information can be used to reconstruct an unlearned model. Following this approach, FedEraser (Liu et al., 2021) enhances model reconstruction efficiency, while FedRecovery (Zhang et al., 2023) improves indistinguishability between unlearned and retained models. Additionally, knowledge distillation can be employed to recover model performance (Wu et al., 2022). The second category involves gradient or weight manipulation, which aims to remove or impair learned representations associated with the data through perturbation (Gu et al., 2024; Fan et al., 2024; Zhao et al., 2023; Deng et al., 2024) or pruning techniques (Wang et al., 2022; Torkzadehmahani et al., 2024). A latest approach (Khalil et al., 2025) proposes NoT, which utilizes weight negation to induce unlearning while preserving model optimality, thereby enabling rapid recovery. The third category focuses on loss function approximation. The unlearning process can be guided using second-order information, such as the Hessian matrix. Liu et al. (2022) proposed leveraging the Fisher information matrix as an approximation of the Hessian to optimize the unlearning process. The fourth category involves reversing the training process. A representative method is gradient ascent (Halimi et al., 2022). Recently, Pan et al. (2025) addressed challenges related to gradient exploration and directional conflicts by proposing FedOSD.

Overall, these four technical pathways encompass most existing federated unlearning methods. However, current approaches largely overlook the relationship between unlearning data and remaining data, as well as how this relationship impacts the unlearning process. Additionally, several prior studies (Thudi et al., 2022; Zhang et al., 2024) highlight that current unlearning evaluation protocols are insufficient to reliably verify unlearning effectiveness. Thus, federated unlearning requires a fine-grained evaluation framework to rigorously assess its efficacy.

## 3 PRELIMINARIES

### 3.1 FEDERATED LEARNING

Federated learning (McMahan et al., 2017; Konečný et al., 2016; Shokri & Shmatikov, 2015) algorithm $f$ is a distributed learning framework that enables the training of a global model $\Phi_G$ by iteratively aggregating knowledge from multiple distributed $K$ clients $C = \{C_k \mid k \in K\}$ without transferring their local datasets $D = \{D_k \mid k \in K\}$. All local datasets $D_k$ are drawn from the data distribution $\mathbb{D}$. At the core of FL lies an algorithmic process where each client $C_k$ trains a local model $\Phi_k$ on its private dataset $D_k$ and periodically communicates the resulting parameters or gradients to a central server. The server then aggregates these updates with specific aggregation algorithms such as Federated Averaging Aggregation (McMahan et al., 2017), to generate the shared global model $\Phi_G \leftarrow f(D)$, which is subsequently redistributed to the clients for further training.

### 3.2 FEDERATED UNLEARNING

Federated unlearning is a subfield of machine unlearning that satisfies the requirements of privacy regulations and user rights in the distributed environment. It focuses on deleting the data locally and removing the influence of data from the global model and all local models (Liu et al., 2024).

**Definition 1.** *Federated Unlearning. We define that $D = \{D_k \mid k \in K\}$ is the set of all local datasets of clients that includes unlearning datasets $D_u$ and remaining datasets $D_r$. It is worth noting that $D = D_u \cup D_r$. The global model $\Phi_G$ is trained on $D$ with federated learning algorithm $f$ that $\Phi_G \leftarrow f(D)$. Thus, the unlearned global model $\Phi_{G_u}$ should satisfy:*

$$\underset{\Phi_{G_u} \leftarrow A(\Phi_G, D_u)}{M} (\Phi_{G_u}(\mathbb{D})) \simeq \underset{\Phi_{G_r} \leftarrow f(D_r)}{M} (\Phi_{G_r}(\mathbb{D})), \tag{1}$$

*where $A$ is the federated unlearning algorithm, $\Phi_{G_u}$ and $\Phi_{G_r}$ are the unlearned global model and the retrained global model respectively, generated by the unlearning process $\Phi_{G_u} \leftarrow A(\Phi_G, D_u)$ and the federated learning retraining $\Phi_{G_r} \leftarrow f(D_r)$. The $M$ denotes a measurement of the model output distribution, such as accuracy. Moreover, $\mathbb{D}$ is the underlying data distribution that all local dataset $D_k$ are sampled from it. The unlearning dataset $D_u$, the remaining dataset $D_r$, and the test dataset $D_{test}$ are sampled from the distribution $\mathbb{D}$.*

Therefore, we conclude that the goal of federated unlearning is to use $A$ to produce an unlearning model $\Phi_{G_u}$ that has a similar output distribution to the retrained model $\Phi_{G_r}$ on the underlying data distribution $\mathbb{D}$. Furthermore, federated unlearning may involve different unlearning targets that depend on requests. Generally, the participants in FL can request to unlearn examples, classes and clients. **In this paper, we focus on client unlearning.** Therefore, the unlearning dataset is defined as $D_u = \{D_k\}_{k \in K_u}$, while the remaining dataset is given by $D_r = \{D_k\}_{k \in K_r}$. Here, $K_u$ and $K_r$ denote the index sets corresponding to the unlearning clients and the remaining clients, respectively.

## 4 UNDERSTANDING MEMORIZATION IN FEDERATED UNLEARNING

Even if the unlearning dataset and the remaining dataset do not share the same individual examples, they may still contain overlapping learnable information at the informational level as shown in Figure 1. This raises a key question: Should the unlearning process remove the overlapping information? **We argue that overlapping or shared information should not be unlearned**.

### 4.1 DEFINITIONS

Formally, we define the information that learned from the remaining dataset $D_r$ as $F_r$, learned from the unlearning dataset $D_u$ as $F_u$:

$$\begin{aligned} F_r &= E(D_r) \\ F_u &= E(D_u), \end{aligned} \tag{2}$$

where $E(\cdot)$ is a information extracting process.

Next, we define the overlapped part as $F_g$ and the non-overlapping information in the unlearning dataset as $F_m$. These could be represented as:

$$F_g = F_r \cap F_u$$
$$F_m = F_u - F_g. \tag{3}$$

We can explain this from the memorization perspective. Insights from memorization theory (Feldman, 2020; Feldman & Zhang, 2020) suggest that memorization information represents unique, non-shared patterns. Therefore, in our unlearning paradigm, the non-overlapping information $F_m$ could be considered as the memorization information of the unlearning dataset $D_u$. In contrast, the overlapping information $F_g$ reflects shared patterns present in both $D_u$ and $D_r$.

**From this perspective, we formally define $F_m$ as memorization or non-overlapping information, and $F_g$ as shared or overlapping information.**

### 4.2 FEDERATED MEMORIZATION UNLEARNING

Intuitively, we can understand the information contained in $\Phi_G$ and $\Phi_{G_r}$ as follows:

$$E(D_r \cup D_u) = F_u \cup F_r = (F_u - F_g) \cup F_r = F_m \cup F_r,$$
$$E(D_r) = F_r. \tag{4}$$

Therefore, the difference in information between $\Phi_G$ and $\Phi_{G_r}$ is given by:

$$E(D_r \cup D_u) - E(D_r) = F_m. \tag{5}$$

This implies that removing $F_m$ from the original model $\Phi_G$ results in an unlearned model $\Phi_{G_u}$ that contains the same information as the retrained model $\Phi_{G_r}$. Intuitively, considering that both $\Phi_{G_u}$ and $\Phi_{G_r}$ only depend on the information $F_r$ derived from $D_r$, their resulting output distributions should be similar:

$$M(\Phi_{G_u}(\mathbb{D})) \simeq M(\Phi_{G_r}(\mathbb{D})), \tag{6}$$

which is consistent with Definition 1.

Thus, we formally propose our federated memorization unlearning definition:

**Definition 2.** *Federated Memorization Unlearning. Federated memorization unlearning refers to the process of removing specific memorization information $F_m$ in unlearning dataset $D_u$. Given a global model $\Phi_G$, the goal is to obtain an unlearned model $\Phi_{G_u}$ such that the influence of $F_m$ is effectively removed:*

$$\Phi_{G_u} \leftarrow A(\Phi_G, F_m). \tag{7}$$

*Furthermore, the unlearned model $\Phi_{G_u}$ should perform similarly to the retrained model $\Phi_{G_r}$ on the entire data distribution $\mathbb{D}$. This implies that we aim to develop an unlearning algorithm $A$ and an unlearned model $\Phi_{G_u}$ that satisfy:*

$$\underset{\Phi_{G_u} \leftarrow A(\Phi_G, F_m)}{M} (\Phi_{G_u}(\mathbb{D})) \simeq \underset{\Phi_{G_r} \leftarrow f(D_r)}{M} (\Phi_{G_r}(\mathbb{D})), \tag{8}$$

*where $M$ denotes a measurement of output distribution similarity.*

## 5 MEMORIZATION-BASED FEDERATED UNLEARNING METRICS

According to the discussion in Section 4, the memorization perspective discloses that the primary difference between the unlearned model and the original model is that the unlearned model has removed the memorization information associated with the unlearning dataset. Thus, it is essential to verify the presence of memorization information to validate the unlearning results.

However, we find that existing metrics are insufficient for evaluating memorization in unlearning and in some cases, they fail even to assess unlearning itself. Specifically, distance-based metrics cannot distinguish the retrained model from the original model. Moreover, performance metrics on the entire dataset are difficult to confirm finer-grained unlearning. Furthermore, backdoor-based

evaluation is invalid, as forgetting a backdoor trigger does not necessarily imply successful unlearning of the target data. These empirical observations are consistent with prior studies (Thudi et al., 2022; Zhang et al., 2024). Further details are provided in Appendix G.

Therefore, since no effective measurement of memorization exists in the context of federated unlearning, we propose a novel unlearning metric specifically designed to evaluate memorization. **Direct evaluation of the memorization information $F_m$ is challenging. Therefore, we move to the example level and indirectly assess whether the examples with memorization information have indeed been forgotten, to validate the unlearning process.**

Specifically, we develop an evaluation based on the memorization scores (Feldman, 2020). **Grouped Memorization Evaluation (GME)** aims to describe the performance of subgroups categorized by varying memorization scores within the unlearning dataset. GME consists of two steps: 1) Calculating unlearning memorization scores independently and grouping examples based on the scores; 2) Evaluating the performance of each subgroup on the unlearned model. The **Unlearning Memorization Score** of each example $(x_i, y_i)$ in the unlearning dataset $D_u$ can be determined using:

$$\text{mem}(f, D_r, D_u, (x_i, y_i)) = \Pr_{\Phi_G \leftarrow f(D_r \cup D_u)}[\Phi_G(x_i) = y_i] - \frac{1}{J} \sum_{j=1}^{J} \Pr_{\Phi_{G_r}^j \leftarrow f(D_r)}[\Phi_{G_r}^j(x_i) = y_i], \quad (9)$$

where $D_u$ refers to the unlearning dataset, which consists of local datasets from clients who request unlearning and the set $D_r$ denotes the remaining dataset after excluding $D_u$. To mitigate randomness, $J$ retrained models $\Phi_{G_r}$ are generated without $D_u$, and the final measurement reflects the difference in the probability of correct classification, $\Pr[\Phi(x_i) = y_i]$, with and without the presence of $D_u$. **Generally, examples with high memorization scores contain more unique and memorization information**.

Subsequently, the examples in the unlearning dataset $D_u$ can be divided into different subgroups $\{T_p\}$ based on their memorization scores:

$$T_p = \{(x_i, y_i) \in D_u \mid \text{mem}_{(x_i, y_i)} \in (\tau_p, \tau_{p+1}]\}, \quad (10)$$

where $\tau$ is the predefined threshold and $(\tau_p, \tau_{p+1}]$ is the unlearning memorization score section of group $T_p$. In this work, we partition the subgroups by sorting the memorization scores from highest to lowest.

Next, we can apply any metric $M$ (such as model accuracy) to evaluate the unlearning effect across different subgroups, as follows:

$$\Delta M_{T_p} = \big| \underset{\Phi_{G_u} \leftarrow A(\Phi_G, D_u)}{M}(\Phi_{G_u}, T_p) - \underset{\Phi_{G_r} \leftarrow f(D_r)}{M}(\Phi_{G_r}, T_p), \big| \quad (11)$$

where $p$ denotes the index of the memorization subgroup $T_p$ and $\Delta M_{T_p}$ is the unlearning difference on the specific subgroup $T_p$. Following Definition 2, we expect the unlearned model to perform similarly to the retrained model. Therefore, we aim for a minimal $\Delta M_{T_p}$ for each subgroup $T_p$. Evaluating these memorization score subgroups enables a more fine-grained assessment of the unlearning effect. In essence, examples with high memorization scores in $D_u$ correspond to the memorization information $F_m$ in our Definition 2. The common performance metrics can then be applied to these subgroups to more precisely characterize the effect of unlearning.

# 6 FEDERATED UNLEARNING BASED ON MEMORIZATION ERASER

## 6.1 INTUITION

For a well-trained model, the information learned from data is encoded in its parameters. Based on the Definition 2, removing memorization information can achieve unlearning effectively. Therefore, pruning parameters that capture memorization information offers a potential unlearning strategy. This comes the first question: how to locate the memorization information in the parameter space. Previous studies (Torkzadehmahani et al., 2024) have identified important parameters relevant to unlearning dataset, based on large gradients and activations. However, these important parameters may capture both shared information $F_g$ and memorization information $F_m$ in the unlearning dataset $D_u$. In contrast, we hypothesize that redundant parameters with respect to the remaining dataset

$D_r$ appear to retain memorization information unique to the unlearning dataset. The redundant parameters refer to the parameters that are not important with respect to the remaining dataset $D_r$. At the information level, this corresponds to $F_m = (F_r \cup F_u) - F_r$. Evidence from federated backdoor attacks (Zhang et al., 2022) further supports this view, showing that backdoors embedded in redundant parameters persist longer. After pruning, the model $\Phi_{G_u}$ requires post fine-tuning on the remaining dataset to reconstruct the learned representations, as the pruning process directly disrupts the original representational structure.

## 6.2 FedMemEraser

The Federated Memorization Eraser (FedMemEraser) is a federated unlearning method that focuses on eliminating memorization. In general, our approach comprises three stages: locating memorization parameters, resetting memorization parameters, and fine-tuning the network on the remaining dataset.

**Overview**. In the locating stage, we utilize the average gradient updates from the remaining clients to identify redundant parameters with minimal updates. Next, we employ the original initialization method to reset the memorization parameters. Finally, we fine-tune the unlearned model on the remaining clients to restore generalization performance. We present the algorithm in Appendix D.

**Stage 1: Memorization location**. As discussed in intuition, redundant or infrequently updated parameters are prone to retain memorization information. Therefore, we define the set of memorization parameters $\Theta_{um}$ as:

$$\Theta_{um} = \{\theta \in \Phi_G \mid \bar{g}_r(\theta) < \gamma\}, \tag{12}$$

where $\theta$ denotes a parameter in the global model $\Phi_G$, $\bar{g}_r(\theta)$ represents the average gradient update of parameter $\theta$, and $\gamma$ is a predefined gradient threshold. The average gradient update $\bar{g}_r(\theta)$ is computed by aggregating the gradients submitted by the remaining clients, and is given by:

$$\bar{g}_r = \frac{1}{|K_r|} \sum_{k \in K_r} g_k, \tag{13}$$

where $K_r$ denotes the index set of remaining clients and $g_k$ is the gradient update of client $k$.

In practice, the threshold $\gamma$ is determined based on a predefined percentage $\rho$ of parameters to be reinitialized. Hence, $\rho$ serves as the main hyperparameter in controlling unlearning.

**Stage 2: Memorization parameters re-initialization**. Since memorization information may not be completely removed by some optimization technologies, we directly choose to reset the memorization parameters. Rather than setting these parameters to zero, we reinitialize them using the original parameter initialization strategy. Specifically, we apply Kaiming Uniform Initialization He et al. (2015) to the convolutional layers and the linear layers.

**Stage 3: Fine-tuning on the remaining dataset**. The final fine-tuning stage follows the standard training procedure, except that only the remaining clients participate. At the $t'$-th round of fine-tuning, the model aggregation is expressed as:

$$\Phi_{G_u}^{t'} = \frac{1}{|K_r|} \sum_{k \in K_r} \Phi_k^{t'}, \tag{14}$$

where $\Phi_k^{t'}$ denotes the local model of client $k$ at round $t'$, and $K_r$ is the index set of remaining clients.

The final global model $\Phi_{G_u}$ is the unlearned model, which effectively removes the influence of the unlearned clients while preserving the generalization performance.

## 7 Experiment

### 7.1 Experiment Setup

**Dataset and Model Architecture**. We evaluate our methods on three benchmark datasets: CIFAR-100, CIFAR-10 (Krizhevsky et al., 2009), EMNIST (Cohen et al., 2017). For CIFAR-100 and

CIFAR-10, we employ a ResNet-34 model and ResNet-18 model (He et al., 2016). EMNIST is processed using a VGG-9 model (Simonyan & Zisserman, 2014).

**Baselines**. During the evaluation, we select Halimi et al. (Halimi et al., 2022), Liu et al. (Liu et al., 2022), FedRecovery (Zhang et al., 2023), FedAU (Gu et al., 2024), and FedOSD (Pan et al., 2025) as the baselines. More detailed information can be found in Appendix E.1.

**Evaluation Methods**. We evaluate federated unlearning along four key dimensions: 1) **Unlearning Efficacy:** we employ Grouped Memorization Evaluation as discussed in Section 5 to quantify the effectiveness of unlearning; 2) **Generalization Performance:** we measure model test accuracy to assess generalization performance; 3) **Local Fairness:** we apply local fairness (Shao et al., 2024) as the degree to which the utility changes of remaining clients deviate from their average after unlearning, where lower variance indicates fairer outcomes. 4) **Time Analysis:** we examine how test performance changes during unlearning and compare it with the trajectory observed under full retraining along the time dimension. We provide specific explanation in Appendix E.2.

**Implementation Details**. We adopt the FedAvg framework with 10 clients. Our experiments focus on the case where a single client requests unlearning, under both IID and Non-IID settings. For the Non-IID case, we follow previous work (Gao et al., 2024; Su & Li, 2023) and partition data using a Dirichlet distribution with concentration parameter $\alpha = 0.5$. More details of the implementation can be found in the Appendix E.3.

## 7.2 EVALUATION RESULTS

**Unlearning Performance**. Table 1 presents the performance of the unlearned model $\Phi_{G_u}$, produced by different unlearning algorithms, across various memorization subgroups within the local dataset $D_u$ of the unlearning client. Basically, this table directly demonstrate that our method most closely approximates the performance of the retrained model, particularly within the subgroup with the highest memorization scores. For example, in the evaluation of CIFAR-100 under Non-IID conditions, the classification accuracies of the retrained model $\Phi_{G_r}$ across the five memorization subgroups are 21.74%, 22.90%, 26.09%, 29.13%, and 65.28%, respectively. In comparison, our method achieves accuracies of 27.25%, 27.39%, 29.71%, 30.43%, and 63.44% on the same subgroups. This highlights the underlying consistent classification patterns shared by our model and the retrained baseline. However, other unlearning baselines fall significantly short of the performance of the retrained model, particularly in high-memorization groups.

**Generalization Performance**. Table 1 also reports the generalization performance across datasets and distribution conditions. Across all scenarios, our method consistently achieves the highest test accuracy on $D_{test}$, demonstrating strong generalization. In CIFAR-100 IID, it reaches 63.33%, far surpassing all baselines. Similar trends appear in other settings, where our method outperforms other unlearning baselines and approaches the retrained baseline generalization performance. These results support Section 4: by removing only memorization while preserving overlapping information, our method enables effective unlearning without harming generalization.

**Local Fairness**. Local fairness measures the variance in loss changes across the remaining local datasets compared to their average after unlearning. A lower variance indicates that the unlearning process induces similar performance changes for all remaining clients, thereby ensuring unlearning fairness. Table 1 shows that our method achieves the highest fairness in several scenarios because it avoids deleting overlapping information, thereby preventing significant performance degradation for clients providing such overlapping information. Specifically, for CIFAR-10, we achieve fairness scores of $0.91 \times 10^{-3}$ and $44.12 \times 10^{-3}$ under IID and Non-IID conditions, respectively, compared to the retrained baselines $0.09 \times 10^{-3}$ and $30.78 \times 10^{-3}$, also outperforming other baselines.

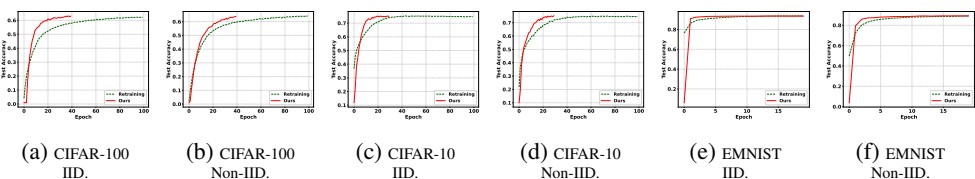

|  |  |  |  |  |  |
|:---:|:---:|:---:|:---:|:---:|:---:|
| (a) CIFAR-100 IID. | (b) CIFAR-100 Non-IID. | (c) CIFAR-10 IID. | (d) CIFAR-10 Non-IID. | (e) EMNIST IID. | (f) EMNIST Non-IID. |

Figure 2: Efficiency comparison between unlearning and retraining.

Table 1: Performance comparison of FedMemEraser vs. retrained and other baseline models

| | Accuracy by Unlearning Memorization Score Subgroup (%) ($\Delta \downarrow$) | | | | | Unlearning Dataset Acc. (%) ($\Delta \downarrow$) | Test Dataset Acc. (%) $\uparrow$ | Local Fairness ($10^{-3}$) $\downarrow$ |
|---|---|---|---|---|---|---|---|---|
| | Group (95%,100%] | Group (90%,95%] | Group (85%,90%] | Group (80%,85%] | Group (0%,80%] | | | |
| **CIFAR-100 IID** | | | | | | | | |
| Halimi et al. | 51.47 ±0.19 (28.27) | 54.93 ±0.19 (27.86) | 52.93 ±0.19 (16.53) | 55.20 ±0.00 (7.60) | 82.77 ±0.11 (8.37) | 78.89 ±0.09 (2.48) | 52.56 ±0.05 | 23.83 ±0.08 |
| Liu et al. | 47.47 ±0.68 (24.27) | 43.60 ±2.14 (16.53) | 50.80 ±1.99 (14.40) | 55.07 ±2.12 (7.47) | 87.52 ±1.09 (3.62) | 81.99 ±1.08 (0.62) | 57.17 ±0.97 | 7.41 ±1.53 |
| FedRecovery | 60.67 ±3.94 (37.47) | 61.60 ±2.14 (34.53) | 57.87 ±0.68 (21.47) | 58.40 ±1.18 (10.80) | 86.70 ±2.58 (4.44) | 83.22 ±0.58 (1.85) | 55.01 ±0.23 | 7.83 ±2.39 |
| FedAU | 59.87 ±1.47 (36.67) | 59.73 ±1.51 (32.66) | 62.53 ±1.91 (26.13) | 60.13 ±3.79 (12.53) | 86.60 ±1.34 (4.54) | 83.04 ±0.97 (1.67) | 53.15 ±0.02 | 7.26 ±1.09 |
| FedOSD | 54.00 ±1.18 (30.80) | 47.87 ±1.32 (20.80) | 52.53 ±3.35 (16.13) | 57.20 ±2.04 (9.60) | 85.72 ±0.50 (5.42) | 81.21 ±0.51 (**0.16**) | 58.55 ±0.14 | 7.67 ±0.21 |
| NoT | 54.80 ±2.04 (31.60) | 55.33 ±1.36 (28.26) | 53.20 ±1.31 (16.80) | 50.93 ±1.15 (3.33) | 83.18 ±0.99 (7.96) | 79.79 ±1.15 (1.58) | 54.72 ±0.49 | 10.85 ±0.94 |
| Ours | 30.80 ±0.86 (**7.60**) | 28.67 ±1.64 (**1.60**) | 39.20 ±1.73 (**2.80**) | 48.53 ±2.22 (**0.93**) | 89.47 ±0.27 (**1.67**) | 80.80 ±0.24 (0.57) | **63.33** ±0.31 | **5.20** ±1.36 |
| Retrained Baseline | 23.20 ±9.62 | 27.07 ±5.85 | 36.40 ±2.90 | 47.60 ±1.18 | 91.14 ±0.68 | 81.37 ±0.51 | 62.29 ±0.58 | 6.46 ±1.81 |
| **CIFAR-100 Non-IID** | | | | | | | | |
| Halimi et al. | 50.87 ±5.71 (29.13) | 43.04 ±5.91 (20.14) | 45.80 ±3.69 (19.71) | 46.23 ±4.85 (17.10) | 64.16 ±3.52 (**1.12**) | 64.08 ±3.80 (2.78) | 61.92 ±1.17 | 87.06 ±13.33 |
| Liu et al. | 43.33 ±0.41 (21.59) | 36.09 ±1.55 (13.19) | 34.35 ±0.35 (8.26) | 33.48 ±1.42 (4.35) | 64.07 ±0.85 (1.21) | 61.73 ±0.92 (0.43) | 58.02 ±0.55 | 54.98 ±8.93 |
| FedRecovery | 42.90 ±3.02 (21.16) | 42.17 ±3.09 (19.27) | 42.61 ±5.64 (16.52) | 42.17 ±3.60 (13.04) | 61.72 ±3.32 (3.56) | 60.93 ±3.12 (0.37) | 48.02 ±0.69 | 110.96 ±21.35 |
| FedAU | 52.46 ±1.96 (30.72) | 47.54 ±1.79 (24.64) | 41.30 ±2.22 (15.21) | 39.42 ±5.96 (10.29) | 57.43 ±2.57 (7.85) | 56.75 ±2.07 (4.55) | 46.68 ±0.97 | 71.90 ±13.28 |
| FedOSD | 56.09 ±4.43 (34.35) | 43.19 ±3.69 (20.29) | 42.46 ±4.41 (16.37) | 38.26 ±3.09 (9.13) | 62.30 ±2.42 (2.98) | 61.53 ±2.47 (**0.23**) | 59.59 ±0.79 | 88.11 ±32.91 |
| NoT | 46.96 ±2.13 (25.22) | 40.58 ±0.82 (17.68) | 45.94 ±0.74 (19.85) | 40.14 ±1.75 (11.01) | 64.74 ±0.10 (**0.54**) | 63.40 ±0.13 (2.10) | 57.06 ±0.16 | 50.08 ±11.57 |
| Ours | 27.25 ±1.25 (**5.51**) | 27.39 ±2.22 (**4.49**) | 29.71 ±1.68 (**3.62**) | 30.43 ±2.48 (**1.30**) | 63.44 ±0.69 (1.84) | 60.37 ±0.74 (0.93) | **62.99** ±0.53 | **42.78** ±6.56 |
| Retrained Baseline | 21.74 ±9.33 | 22.90 ±8.58 | 26.09 ±4.00 | 29.13 ±1.28 | 65.28 ±1.22 | 61.30 ±0.26 | 63.89 ±0.28 | 23.92 ±5.25 |
| **CIFAR-10 IID** | | | | | | | | |
| Halimi et al. | 58.27 ±0.19 (45.47) | 82.27 ±0.19 (47.87) | 92.13 ±0.38 (14.53) | 94.00 ±0.33 (1.87) | 97.84 ±0.07 (1.28) | 94.89 ±0.04 (4.58) | 74.72 ±0.09 | 6.74 ±0.05 |
| Liu et al. | 53.33 ±1.80 (40.53) | 70.00 ±1.13 (35.60) | 76.53 ±1.86 (**1.07**) | 87.07 ±0.94 (5.06) | 98.33 ±0.24 (0.79) | 93.16 ±0.25 (2.85) | 68.13 ±0.24 | 1.23 ±0.25 |
| FedRecovery | 73.87 ±4.76 (61.07) | 85.47 ±3.30 (51.07) | 91.87 ±3.60 (14.27) | 94.27 ±1.64 (2.14) | 95.55 ±1.35 (1.53) | 70.81 ±0.82 | 8.73 ±2.47 | |
| FedAU | 77.87 ±6.71 (65.07) | 78.67 ±6.88 (44.27) | 82.93 ±2.17 (5.33) | 83.60 ±3.22 (8.53) | 91.67 ±2.53 (7.45) | 89.09 ±1.78 (1.22) | 62.48 ±1.20 | 6.04 ±1.76 |
| FedOSD | 66.80 ±4.09 (54.00) | 83.47 ±1.47 (49.07) | 91.87 ±0.38 (14.27) | 91.07 ±2.00 (**1.06**) | 97.05 ±1.03 (2.07) | 94.51 ±1.00 (4.20) | 70.39 ±0.34 | 6.27 ±0.69 |
| NoT | 36.40 ±1.70 (23.60) | 66.27 ±0.19 (31.87) | 82.00 ±1.31 (4.40) | 85.20 ±1.42 (6.93) | 96.22 ±0.87 (2.90) | 91.06 ±0.75 (0.75) | 74.56 ±0.28 | 6.16 ±0.49 |
| Ours | 20.27 ±1.54 (**7.47**) | 50.40 ±3.44 (**16.00**) | 70.93 ±2.29 (6.67) | 85.60 ±0.86 (6.53) | 98.66 ±0.20 (**0.46**) | 90.56 ±0.35 (**0.25**) | **74.97** ±0.27 | **0.91** ±0.07 |
| Retrained Baseline | 12.80 ±9.34 | 34.40 ±19.80 | 77.60 ±14.71 | 92.13 ±5.58 | 99.12 ±0.62 | 90.31 ±0.16 | 74.79 ±0.38 | 0.09 ±0.01 |
| **CIFAR-10 Non-IID** | | | | | | | | |
| Halimi et al. | 45.02 ±0.44 (32.87) | 44.44 ±0.44 (23.78) | 45.33 ±0.66 (19.78) | 58.06 ±0.89 (15.65) | 84.27 ±0.18 (7.89) | 79.16 ±0.15 (1.25) | 65.49 ±0.13 | 172.45 ±1.63 |
| Liu et al. | 73.68 ±13.07 (61.53) | 61.97 ±13.91 (41.31) | 61.21 ±11.47 (35.66) | 61.35 ±9.64 (18.94) | 84.94 ±5.88 (4.53) | 84.94 ±5.88 (4.53) | 68.22 ±2.71 | 219.91 ±103.08 |
| FedRecovery | 43.61 ±1.17 (31.46) | 42.88 ±3.12 (22.22) | 44.55 ±0.58 (19.00) | 51.33 ±2.31 (8.92) | 81.55 ±1.99 (10.61) | 76.92 ±1.48 (3.49) | 62.45 ±2.14 | 148.91 ±28.94 |
| FedAU | 77.10 ±11.57 (64.95) | 67.45 ±8.10 (46.79) | 69.00 ±4.62 (43.45) | 69.17 ±3.56 (26.76) | 85.24 ±4.83 (6.92) | 82.38 ±5.32 (1.97) | 61.38 ±0.49 | 107.94 ±46.10 |
| FedOSD | 73.68 ±4.19 (61.53) | 61.50 ±1.01 (40.84) | 61.37 ±0.79 (35.82) | 66.67 ±1.01 (24.26) | 94.38 ±0.53 (**2.22**) | 89.18 ±0.43 (8.77) | 73.59 ±0.18 | 80.27 ±1.17 |
| NoT | 30.22 ±0.44 (18.07) | 36.62 ±1.01 (15.96) | 43.15 ±0.58 (17.60) | 55.87 ±1.01 (13.46) | 88.15 ±0.67 (4.01) | 80.74 ±0.45 (0.33) | 72.16 ±0.78 | 76.15 ±12.65 |
| Ours | 23.99 ±2.86 (**11.84**) | 27.86 ±1.81 (**7.20**) | 39.72 ±0.66 (**14.17**) | 47.57 ±2.18 (**5.16**) | 88.80 ±1.17 (3.36) | 80.16 ±0.81 (**0.25**) | **74.43** ±0.78 | **44.12** ±5.20 |
| Retrained Baseline | 12.15 ±8.80 | 20.66 ±14.61 | 25.55 ±18.07 | 42.41 ±10.43 | 92.16 ±4.27 | 80.41 ±0.09 | 74.93 ±0.32 | 30.78 ±6.58 |
| **EMNIST IID** | | | | | | | | |
| Halimi et al. | 66.67 ±0.98 (19.28) | 86.06 ±0.42 (12.16) | 95.56 ±0.34 (4.27) | 97.72 ±0.08 (2.28) | 99.80 ±0.04 (0.20) | 96.67 ±0.11 (0.57) | 92.64 ±0.03 | 2.85 ±0.05 |
| Liu et al. | 62.22 ±1.70 (14.83) | 90.50 ±1.47 (7.72) | 98.83 ±0.00 (1.00) | 99.17 ±0.24 (0.83) | 100.00 ±0.00 (**0.00**) | 97.13 ±0.18 (0.11) | 92.90 ±0.05 | 5.30 ±0.54 |
| FedRecovery | 74.78 ±0.75 (27.39) | 89.17 ±0.76 (9.05) | 96.11 ±0.06 (3.72) | 98.11 ±0.34 (1.89) | 99.77 ±0.10 (0.23) | 97.32 ±0.12 (0.08) | 92.56 ±0.05 | 2.61 ±0.59 |
| FedAU | 69.56 ±3.68 (22.17) | 84.83 ±3.08 (13.39) | 89.00 ±3.09 (10.83) | 93.94 ±1.23 (6.06) | 98.78 ±0.41 (1.22) | 94.21 ±0.81 (3.03) | 88.57 ±0.47 | 3.29 ±0.21 |
| FedOSD | 69.06 ±1.88 (21.67) | 87.67 ±2.13 (10.55) | 96.61 ±1.85 (3.22) | 97.61 ±1.11 (2.39) | 99.87 ±0.12 (0.13) | 96.73 ±0.65 (0.51) | 92.62 ±0.40 | 2.85 ±0.22 |
| NoT | 48.50 ±0.14 (**1.11**) | 75.33 ±1.06 (22.89) | 91.11 ±0.08 (8.72) | 96.67 ±0.00 (3.33) | 99.66 ±0.00 (0.34) | 94.45 ±0.05 (2.79) | 92.62 ±0.04 | 4.04 ±0.03 |
| Ours | 53.06 ±1.86 (**5.67**) | 94.33 ±0.85 (**3.89**) | 99.50 ±0.14 (**0.33**) | 99.83 ±0.00 (**0.17**) | 99.97 ±0.02 (0.03) | 97.19 ±0.11 (**0.05**) | **93.50** ±0.06 | **0.70** ±0.15 |
| Retrained Baseline | 47.39 ±1.95 | 98.22 ±1.23 | 99.83 ±0.14 | 100.00 ±0.00 | 100.00 ±0.00 | 97.24 ±0.01 | 93.60 ±0.01 | 0.18 ±0.03 |
| **EMNIST Non-IID** | | | | | | | | |
| Halimi et al. | 58.13 ±0.20 (37.56) | 64.69 ±0.15 (2.22) | 79.33 ±0.21 (8.63) | 94.78 ±0.06 (2.30) | 99.25 ±0.01 (0.58) | 93.48 ±0.02 (0.50) | 88.82 ±0.01 | 45.02 ±1.24 |
| Liu et al. | 49.88 ±6.62 (29.31) | 64.04 ±6.05 (2.87) | 84.87 ±3.43 (3.09) | 95.93 ±1.57 (1.15) | 99.76 ±0.11 (0.07) | 93.56 ±1.04 (0.58) | 88.75 ±0.46 | 136.33 ±41.18 |
| FedRecovery | 67.00 ±9.32 (46.43) | 77.64 ±9.97 (10.73) | 87.92 ±4.45 (**0.04**) | 95.68 ±0.87 (1.40) | 99.62 ±0.15 (0.21) | 95.60 ±1.62 (2.62) | 87.80 ±0.57 | 46.23 ±20.40 |
| FedAU | 69.95 ±7.48 (49.38) | 67.37 ±6.25 (**0.46**) | 84.55 ±4.80 (3.41) | 95.97 ±2.06 (1.11) | 99.78 ±0.09 (0.05) | 94.43 ±1.43 (1.45) | 87.60 ±0.98 | 31.94 ±16.40 |
| FedOSD | 57.55 ±7.34 (36.98) | 64.98 ±6.20 (1.93) | 85.49 ±3.92 (2.47) | 96.22 ±1.62 (0.86) | 99.80 ±0.03 (0.03) | 93.93 ±1.18 (0.95) | 88.99 ±0.40 | 33.85 ±12.85 |
| NoT | 38.88 ±0.25 (18.31) | 65.23 ±0.17 (1.68) | 86.07 ±0.70 (1.89) | 95.77 ±0.15 (1.31) | 99.80 ±0.01 (0.03) | 93.37 ±0.04 (0.39) | 89.30 ±0.03 | 22.64 ±3.81 |
| Ours | 33.33 ±0.76 (**12.76**) | 63.50 ±1.43 (3.41) | 85.45 ±1.22 (2.51) | 96.59 ±0.06 (0.49) | 99.80 ±0.01 (0.03) | 93.16 ±0.24 (0.18) | **89.21** ±0.10 | **19.35** ±1.41 |
| Retrained Baseline | 20.57 ±13.60 | 66.91 ±6.60 | 87.96 ±2.41 | 97.08 ±0.71 | 99.83 ±0.04 | 92.98 ±0.17 | 89.18 ±0.06 | 18.73 ±2.73 |

We report accuracies of the unlearned model $\Phi_{G_u}$ on memorization subgroups $T_p$ of the unlearning dataset $D_u$ and the test dataset $D_{\text{test}}$, along with the local fairness metric. Subgroups are defined by percentile ranges of unlearning memorization scores (e.g., (95%, 100%] contains the top 5% memorization examples). Accuracy differences $\Delta$ from retrained baselines are shown in blue, with the smallest differences and the best test, fairness performance highlighted in red.

**Time Analysis**. Figure 2 illustrates the test accuracy dynamics in training epochs for both retraining and our proposed unlearning method. Compared to retraining, our unlearning method achieves the highest test accuracy in significantly less time, showing around 50% time improvement. This is especially notable in the CIFAR-100 IID scenario (Figure F4a), where our method rapidly converges in 20 epochs, while retraining methods take nearly four times longer to stabilize. These practical results illustrate the time efficiency of our method. For other baselines applying different methods, we qualitatively analyze time complexity. FedRecovery (Zhang et al., 2023) is the fastest because it utilize historical data, trading space for time. In the case of FedAU (Gu et al., 2024), it retrains only the final classification layer, which requires minimal computation time. In contrast, the method proposed by Liu et al. (2022) employs a complex optimization procedure, demanding more time. Finally, both the methods of Halimi et al. (2022) and FedOSD (Pan et al., 2025) include a post-training stage, resulting in a similar time consumption compared to our approach.

## 7.3 ABLATION STUDIES

We conduct ablation experiments to analyze the factors influencing FedMemEraser.

**Effect of Pruning Ratio**. Fine-tuning alone ($\rho = 0\%$) fails to unlearn, leaving large accuracy gaps in high-memorization groups. A moderate pruning ratio ($\rho = 40\%$) achieves performance close to retraining with a 50% time improvement. The high pruning ($\rho \geq 60\%$) may cause excessive forgetting and converge toward retraining time and behavior. More details have been provided in Appendix F.1.

**Effect of Data Distribution**. Our method remains robust across diverse Non-IID settings, matching retraining performance. Notably, only fine-tuning suffices for unlearning when no information overlaps between unlearning and remaining dataset. See Appendix F.2 for details.

**Impact of Parameter Selection Strategy**. We compare our redundant parameter selection strategy with other parameter selection strategies (Fan et al., 2024; Torkzadehmahani et al., 2024). The results indicate that our strategy achieves a better balance between generalization performance and unlearning performance. Detailed results are provided in Appendix F.3.

**Impact of Unlearning Multiple Clients**. We demonstrate that our method remains effective when multiple clients request unlearning. We provide detailed results in Appendix F.4.

**Impact of Large Scale FL Setting**. We demonstrate that our method performs well in federated learning scenarios with a larger number of clients. Detailed results are provided in Appendix F.5.

### 7.4 EXTRA DISCUSSION

**Relationship between Parameters and Information**. We attempt to investigate the relationship between parameters and information representation. Our experiments show that important parameters mainly encode shared/generalized information, whereas redundant parameters are more closely tied to memorization. The results are presented in Appendix H.1.

**Fast Memorization Proxy**. Evaluating memorization scores typically incurs substantial computational cost. To mitigate this issue, a recent study (Ravikumar et al., 2025) demonstrates that the cumulative sample loss (CSL) can serve as an efficient proxy to replace the memorization score. CSL quantifies the total loss a model accumulates on each sample throughout the training process. Thus, it does not require any additional computation. Further details are provided in Appendix H.2.

**Pruning Ratio Selection**. The pruning ratio $\rho$ is a key hyperparameter in our method. Accordingly, we propose a method to select the pruning ratio $\rho$ based on the proportion of redundant parameters. Given the sparsity of parameters in neural networks, it is feasible to determine a threshold of small magnitude based on parameter importance to achieve the desired pruning ratio. Further details are provided in Appendix H.3.

**Extending to Class-Level and Example-Level Unlearning**. We evaluate our approach on both class-level and example-level unlearning tasks. Our method can be seamlessly applied to class-level unlearning and demonstrates strong performance in this setting. For example-level unlearning, the system may accumulate multiple single-example unlearning requests before executing the unlearning pipeline. Therefore, example-level unlearning is similar to client-level unlearning. Our approach continues to perform effectively in this setting. A detailed discussion is provided in Appendix H.4.

**Loss surface of the shortest unlearning path**. We examine the loss surface of the shortest unlearning path. The analysis reveals that the original and retrained models lie in different loss basins. Even small perturbations to the original parameters sharply increase loss. This implies that unlearning is inherently closer to a partial re-training process, highlighting the necessity of performance recovery. We provided details in Appendix H.5.

## 8 CONCLUSION

By distinguishing between shared and memorized information, we argue that effective unlearning should target only memorization information, preserving model generalization and the contributions of other clients. To this end, we introduced FedMemEraser, a novel approach that selectively removes memorization parameters, achieving robust and efficient unlearning. Our findings offer both theoretical insights and practical strategies for strengthening federated unlearning.

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

## APPENDIX A    STATEMENT ON THE USE OF LARGE LANGUAGE MODELS (LLMs)

The authors confirm that large language models were used only for text improvement purposes (including grammar, clarity, and stylistic refinement). No part of the conceptual development, experimental design, analysis, or substantive content of the paper relied on LLM assistance. All scientific contributions are entirely the work of the authors.

## APPENDIX B    REPRODUCIBILITY STATEMENT

For reproducibility, we provide detailed descriptions of datasets, model architectures in the main text. Moreover, we provide the algorithm in Appendix D and implement details in Appendix E.3. With these, our method and experiment results can be easily reproduced. Upon acceptance, we will release our code and instructions to fully reproduce our results.

## APPENDIX C    NOTATIONS

Table C1: Table of notations.

| Symbol | Description |
| --- | --- |
| $K$ | The set of client indices. |
| $K_u$ | The set of unlearning client indices. |
| $K_r$ | The set of remaining client indices. |
| $C_k$ | The client corresponding to index $k$. |
| $\mathbb{D}$ | The underlying data distribution. |
| $D_k$ | The local private dataset of the client with index $k$. |
| $D$ | The set of all local datasets from all clients. |
| $D_u$ | The set of all local datasets from unlearning clients with indices in $K_u$. |
| $D_r$ | The set of all local datasets from remaining clients with indices in $K_r$. |
| $D_{test}$ | The test dataset that simulates the unseen data. |
| $f$ | The federated learning algorithm. |
| $A$ | The federated unlearning algorithm. |
| $\Phi_G$ | The global model. |
| $\Phi_{G_u}$ | The global unlearned model. |
| $\Phi_{G_r}$ | The global retrained model. |
| $M$ | The performance measurement method, e.g., accuracy. |
| $F_g(\Phi)$ | The overlapped information that $\Phi$ learns from both $D_u$ and $D_r$. |
| $F_m(\Phi)$ | The non-overlapping information that $\Phi$ only learn from $D_u$. |
| $\tau$ | The pre-defined threshold to split memorization score groups. |
| $T_p$ | The group of examples categorized by the memorization score at index $p$. |
| $\theta$ | The partial parameters in model $\Phi$. |
| $g_k$ | The gradient update of client $C_k$ on its local dataset $D_k$. |
| $\gamma$ | The predefined gradient threshold. |
| $\rho$ | The predefined percentage gradient threshold. |
| $\alpha$ | The concentration parameter of a Dirichlet distribution. |

# APPENDIX D    ALGORITHM DESCRIPTION

In this subsection, we give the completed pipeline of our unlearning method in Algorithm 1.

---

**Algorithm 1** FedMemEraser

---

**Input:** Trained global model $\Phi_G$; remaining clients $\{C_k \mid k \in K_r\}$ and its index set $K_r$; drop ratio $\rho$ (or threshold $\gamma$); local learning rate $\lambda$.

**Output:** Unlearned model $\Phi_G^u$.

    ▷ Stage 1: Locate memorization parameters.

1: Server broadcasts $\Phi_G$ to all remaining client $\{C_k \mid k \in K_r\}$.

2: Initialize local model $\Phi_k \leftarrow \Phi_G$ for each client $C_k$ that $k \in K_r$.

3: Calculate gradient update

$$g_k = \nabla_{\Phi_k}\mathcal{L}(\Phi_k, D_k)$$

    for each client $C_k$ in $\{C_k \mid k \in K_r\}$.

4: Compute average gradient of remaining clients $\{C_k \mid k \in K_r\}$:

$$\bar{g}_r = \frac{1}{|K_r|}\sum_{k \in K_r} g_k$$

5: Select *redundant parameters*:

$$\Theta_{um} = \{\theta \in \Phi_G \mid \bar{g}_r(\theta) < \gamma\},$$

    **or** equivalently choose the lowest $\rho$ parameters by $|g_r(\theta)|$.

    ▷ Stage 2: Reset memorization parameters.

6: Re-initialize $\Theta_{um}$ using the original scheme (e.g., Kaiming Uniform for conv/linear) and denote the reset model as $\Phi_{G_u}^0$.

    ▷ Stage 3: Fine-tuning on remaining clients.

7: **for** $t \leftarrow 0$ to $t_{end}$ **do**

8:     Server broadcasts $\Phi_{G_u}^t$ to remaining clients $\{C_k \mid k \in K_r\}$.

9:     **for all** $k \in K_r$ **in parallel do**

10:        $\Phi_k^{r_0} \leftarrow \Phi_k^t$

11:        **for** $r \leftarrow 0$ to $r_{end}$ **do**

12:           Update local model

$$\Phi_k^{r+1} = \Phi_k^r - \lambda\nabla_{\Phi_k^r}\mathcal{L}(\Phi_k^r, D_k).$$

13:        **end for**

14:        $\Phi_k^{t+1} \leftarrow \Phi_k^{r_{end}}$

15:     **end for**

16:     Compute average model of remaining clients $\{C_k \mid k \in K_r\}$:

$$\Phi_{G_u}^{t+1} = \frac{1}{|K_r|}\sum_{k \in K_r} \Phi_k^{t+1}$$

17: **end for**

18: $\Phi_{G_u} \leftarrow \Phi_{G_u}^{t_{end}}$

19: **return** $\Phi_{G_u}$

---

## APPENDIX E  EXPERIMENT DETAILS

### APPENDIX E.1  BASELINES

In this subsection, we provide a detailed explanation of the unlearning baselines:

- **Halimi et al.** (Halimi et al., 2022) This work attempts to revert the training process via gradient ascent. However, since gradient ascent can lead to an arbitrary model, the unlearned model is restricted within a predefined threshold in terms of the distance $l_2$.
- **Liu et al.** (Liu et al., 2022) Liu et al. propose a method that leverages the Fisher Information Matrix (FIM) to estimate the inverse Hessian matrix and guide the unlearning process.
- **FedRecovery.** (Zhang et al., 2023) FedRecovery aims to reduce the updates related to unlearning data by utilizing historical update storage, and applies differential privacy to enhance indistinguishability compared to the retrained model.
- **FedAU.** (Gu et al., 2024) FedAU perturbs the learned features of unlearning data by employing random labels. Through linear operations, retraining the classification layer is a more efficient unlearning method.
- **FedOSD.** (Pan et al., 2025) FedOSD introduces advanced techniques to address gradient exploration during gradient ascent and gradient conflicts during unlearning, thereby mitigating model utility degradation.
- **NoT.** (Khalil et al., 2025) NoT is a recent federated unlearning approach that aims to apply strong and resilient weight negation to model weights in order to induce unlearning and enable rapid recovery.

### APPENDIX E.2  EVALUATION METHODS

**Unlearning Efficacy**. We employ Grouped Memorization Evaluation, as discussed in Section 6, to quantify the effectiveness of unlearning. In this experiment, we set $J = 3$ to mitigate randomness. Moreover, we divide the examples according to their memorization score percentiles: $(95\%, 100\%]$, $(90\%, 95\%]$, $(85\%, 90\%]$, $(80\%, 85\%]$, and $(0\%, 80\%]$. According to the long-tail theory (Feldman, 2020; Feldman & Zhang, 2020), only the tail portion corresponds to memorization examples. Consequently, we regard the top 20% of examples by memorization score, i.e., those in the $(80\%, 100\%]$ range, as representing memorization information. The remaining examples, in the range $(0\%, 80\%]$, are considered mainly carrying shared or generalizable information. To assess unlearning effectiveness, we compare the performance of the unlearned model to that of a retrained model. A smaller performance difference indicates more effective unlearning, particularly in terms of removing memorization information.

**Generalization Performance**. Generalization performance is typically evaluated using a test dataset. Accordingly, we use

$$\text{Accuracy}(\Phi_{G_u}, D_{test})$$

to assess the generalization capability of the model.

**Local Fairness**. Local fairness (Shao et al., 2024) in the context of federated unlearning is defined as the extent to which the utility changes of the remaining clients deviate from their average after unlearning. This definition ensures that the unlearning process induces similar performance variation across all remaining clients, thereby promoting fairness. The performance change for a client $C_k$ is given by

$$\Delta L_k(\Phi_{G_u}) = L(\Phi_{G_u}, D_k) - L(\Phi_G, D_k).$$

The local fairness metric can then be expressed as

$$\text{LocalFairness}(\Phi_{G_u}) = \sum_{k \in K_r} \left| \Delta L_k(\Phi_{G_u}) - \overline{\Delta L} \right|,$$

where $\overline{\Delta L}$ is the average performance change across all remaining clients.

**Time Analysis**. We record the changes in generalization performance throughout the unlearning process and compare them with the performance trajectory observed during full retraining, with

respect to the time dimension. More specifically, we compare the time taken by the unlearned model and the retrained model to reach their respective peak generalization performance. We then report how many times faster our unlearning approach is compared to the retrained baseline.

### APPENDIX E.3   IMPLEMENTATION DETAILS

For the federated learning experiments, we adopt the Adam optimizer with a learning rate of $1 \times 10^{-4}$, and each client performs 3 local training epochs per round. Our method is fine-tuned for 40, 30, and 20 rounds after pruning on CIFAR-100, CIFAR-10, and EMNIST, respectively.

Regarding pruning, we set the pruning ratios to 0.3 and 0.4 for CIFAR-10 under IID and Non-IID conditions, respectively, and apply the same ratios for EMNIST. For CIFAR-100, we employ data augmentation techniques to enhance training, which effectively compresses information into the most important neurons while leaving more redundant parameters. Consequently, we adopt higher pruning ratios of 0.8 and 0.85 for CIFAR-100 under IID and Non-IID settings.

## APPENDIX F   ABLATION STUDIES

In this section, we examine how the pruning ratio, data distribution, parameter selection, and the number of clients affect the performance of our unlearning algorithms.

### APPENDIX F.1   EFFECT OF PRUNING RATIO

In this subsection, we demonstrate how the pruning ratio $\rho$ influences the unlearning performance. As presented in Section 6, the hyperparameter $\rho$ determines how many parameters of the model are re-initialized during unlearning.

Table F2: Performance of FedMemEraser under different pruning ratios

| Pruning Ratio | Accuracy by Unlearning Memorization Score Subgroup (%) ($\Delta \downarrow$) | | | | | Unlearning Dataset Acc. (%) ($\Delta \downarrow$) | Test Dataset Acc. (%) $\uparrow$ | Local Fairness ($10^{-3}$) $\downarrow$ |
|---|---|---|---|---|---|---|---|---|
| | Group (95%,100%] | Group (90%,95%] | Group (85%,90%] | Group (80%,85%] | Group (0%,80%] | | | |
| 0 | 86.60 ±0.22 (70.71) | 85.60 ±0.59 (54.61) | 84.74 ±0.88 (46.89) | 87.17 ±0.59 (36.47) | 98.50 ±0.21 (9.12) | 96.30 ±0.10 (15.89) | 77.37 ±0.22 | 49.37 ±2.09 |
| 0.2 | 43.30 ±1.23 (27.41) | 49.92 ±4.06 (18.93) | 56.70 ±1.59 (18.85) | 68.39 ±1.89 (17.69) | 94.12 ±0.07 (4.74) | 87.39 ±0.30 (6.98) | 77.06 ±0.06 | 31.49 ±3.13 |
| 0.4 | 25.55 ±2.24 (9.66) | 29.42 ±1.17 (**1.57**) | 40.65 ±1.75 (2.80) | 48.36 ±2.89 (**2.34**) | 89.00 ±0.74 (**0.38**) | 80.72 ±0.44 (**0.31**) | **75.36** ±0.35 | 32.55 ±6.56 |
| 0.6 | 18.22 ±1.53 (**2.33**) | 27.54 ±2.72 (3.45) | 33.49 ±1.54 (4.36) | 44.44 ±0.59 (6.26) | 87.58 ±0.54 (1.80) | 78.72 ±0.31 (1.69) | 73.47 ±0.51 | **30.50** ±3.46 |
| 0.8 | 19.00 ±0.58 (3.11) | 25.04 ±3.26 (5.95) | 35.20 ±2.10 (2.65) | 46.32 ±1.35 (4.38) | 88.50 ±0.94 (0.85) | 79.48 ±0.72 (0.93) | 74.24 ±0.64 | 40.59 ±4.77 |
| 1 | 19.16 ±0.66 (3.27) | 25.98 ±4.03 (5.01) | 38.01 ±0.88 (**0.16**) | 43.82 ±1.55 (6.88) | 88.50 ±0.62 (0.88) | 79.55 ±0.34 (0.86) | 74.06 ±0.82 | 36.28 ±0.90 |
| Retrained Baseline | 15.89 ±8.80 | 30.99 ±14.61 | 37.85 ±18.07 | 50.70 ±10.43 | 89.38 ±4.27 | 80.41 ±0.09 | 74.93 ±0.32 | 30.78 ±6.58 |

This table represents the accuracies of the unlearned model $\Phi_{G_u}$ on the unlearning dataset $D_u$, its subgroups $\{T_p\}$ split by memorization scores and test dataset $D_{test}$, along with the local fairness metric across different pruning ratios. We also denote the difference $\Delta$ between the retrained baselines and the corresponding pruning ratio results in blue, and have highlighted the best values in **red**.

Table F2 illustrates the performance of the unlearned model under different pruning ratios. When $\rho$ is 0%, it corresponds to pure fine-tuning, which leads to unlearning failure, as no memorization information are removed. In this case, the accuracy gap remains very high, reaching 70.71% on the top memorization group. This indicates that fine-tuning alone is insufficient for unlearning. In contrast, when $\rho$ equals 40%, meaning that 40% of the parameters are dropped, the unlearned model $\Phi_{G_u}$ performs similarly to the retrained model $\Phi_{G_r}$ in different subgroups of memorization scores. This suggests that key memorization information in the unlearning dataset $D_u$ have been successfully forgotten, providing evidence for our hypothesis that memorization information

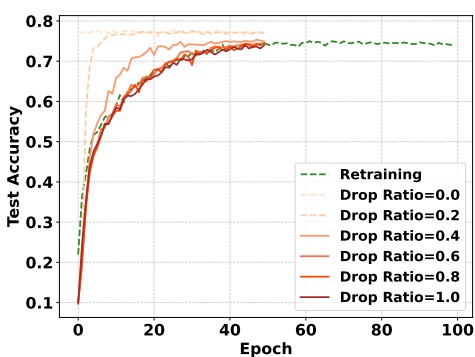

Figure F1: Efficiency comparison across pruning ratios $\rho$

are primarily stored in redundant parameters.

Moreover, the local fairness reaches 32.55, approaching the retrained baseline that ensures client-level fairness in federated unlearning. In the time dimension, the runtime of the proposed approach represents an improvement 50% over the retrained baseline, as shown in Figure F1.

However, a higher pruning ratio $\rho$ leads to excessive forgetting, resulting in accuracies that are noticeably lower than those of the retrained baseline. Additionally, the unlearning times approaching those of full retraining, as shown in Figure F1. Specifically, when $\rho$ exceeds 60%, the unlearning time closely approaches the retraining time. Therefore, selecting an appropriate pruning ratio $\rho$ is crucial. Nevertheless, the optimal value of $\rho$ depends on the distribution of memorization information across the model parameters, which is influenced by factors such as model architecture, data distribution, and the federated learning setting. We discussed how to choose the pruning ratio in Appendix H.3.

## APPENDIX F.2  EFFECT OF DATA DISTRIBUTION INFLUENCE

In this subsection, we examine how data distribution influences federated unlearning. Specifically, we explore how the discrepancy between the remaining dataset and the unlearning dataset impacts the unlearning process.

As discussed in Section 4, the relationship between the remaining dataset and the unlearning dataset is a critical factor in evaluating the difficulty of unlearning. Qualitatively, there are two main scenarios: 1) the remaining dataset and the unlearning dataset exhibit significant overlap, and 2) the two datasets are substantially different. These relationships can be quantitatively described by the degree of the Non-IID data. A higher degree of Non-IID data ($\alpha \downarrow$) implies a smaller overlap between the remaining and unlearning datasets. In an extreme Non-IID scenario ($\alpha \rightarrow 0$), where each client holds data from a single, unique class, unlearning any one client results in minimal overlap. In contrast, IID settings typically involve more shared information, which can result in greater overlap between the two datasets in the context of unlearning.

Table F3: Performance of FedMemEraser across different Non-IID scenarios

| Non-IID Degree | Accuracy by Unlearning Memorization Score Subgroup (%) ($\Delta \downarrow$) | | | | | Unlearning Dataset Acc. (%) ($\Delta \downarrow$) | Test Dataset Acc. (%) ↑ | Local Fairness ($10^{-3}$) ↓ |
|---|---|---|---|---|---|---|---|---|
| | Group (95%,100%] | Group (90%,95%] | Group (85%,90%] | Group (80%,85%] | Group (0%,80%] | | | |
| $\alpha \rightarrow 0$ | | | | | | | | |
| Only Fine-tuning | 0.00 ±0.00 (0.00) | 0.00 ±0.00 (0.00) | 0.00 ±0.00 (0.00) | 0.00 ±0.00 (0.00) | 0.00 ±0.00 (0.00) | 0.00 ±0.00 (0.00) | 16.26 ±1.42 | 63.12 ±3.71 |
| Ours | 0.00 ±0.00 (0.00) | 0.00 ±0.00 (0.00) | 0.00 ±0.00 (0.00) | 0.00 ±0.00 (0.00) | 0.00 ±0.00 (0.00) | 0.00 ±0.00 (0.00) | 12.87 ±1.39 | 67.70 ±11.78 |
| Retrained Baseline | 0.00 ±0.00 | 0.00 ±0.00 | 0.00 ±0.00 | 0.00 ±0.00 | 0.00 ±0.00 | 0.00 ±0.00 | 13.92 ±0.70 | 98.57 ±13.90 |
| $\alpha = 0.1$ | | | | | | | | |
| Only Fine-tuning | 92.79 ±1.46 (44.40) | 89.79 ±0.80 (36.56) | 80.75 ±1.50 (30.49) | 85.53 ±0.97 (31.14) | 52.44 ±1.55 (13.40) | 56.41 ±1.33 (15.11) | 54.54 ±0.16 | 126.43 ±2.51 |
| Ours | 51.35 ±6.21 (2.96) | 54.26 ±6.63 (1.03) | 50.78 ±7.40 (0.52) | 49.61 ±7.99 (4.78) | 37.88 ±2.50 (1.16) | 40.12 ±3.01 (1.18) | 50.26 ±2.44 | 267.25 ±124.71 |
| Retrained Baseline | 48.39 ±17.43 | 53.23 ±16.03 | 50.26 ±11.31 | 54.39 ±6.36 | 39.04 ±1.45 | 41.30 ±2.77 | 51.34 ±1.35 | 110.65 ±45.80 |
| $\alpha = 1$ | | | | | | | | |
| Only Fine-tuning | 100.00 ±0.00 (84.93) | 99.62 ±0.00 (83.47) | 98.98 ±0.18 (72.93) | 95.51 ±0.48 (68.84) | 97.63 ±0.11 (13.36) | 97.73 ±0.10 (23.54) | 76.73 ±0.12 | 10.35 ±0.02 |
| Ours | 25.67 ±2.71 (10.60) | 25.51 ±1.45 (9.36) | 34.10 ±0.54 (8.05) | 37.82 ±3.44 (11.15) | 80.96 ±0.35 (3.31) | 73.26 ±0.33 (0.93) | 76.38 ±0.25 | 9.16 ±0.07 |
| Retrained Baseline | 15.07 ±10.66 | 16.15 ±11.42 | 26.05 ±18.42 | 26.67 ±18.86 | 84.27 ±3.14 | 74.19 ±0.55 | 77.46 ±0.42 | 3.70 ±0.58 |
| $\alpha = 10$ | | | | | | | | |
| Only Fine-tuning | 100.00 ±0.00 (93.00) | 99.85 ±0.22 (86.76) | 99.54 ±0.00 (80.36) | 98.63 ±0.00 (66.67) | 99.63 ±0.01 (7.05) | 99.60 ±0.01 (21.84) | 78.69 ±0.02 | 1.45 ±0.00 |
| Ours | 14.16 ±1.63 (7.16) | 22.68 ±2.28 (9.59) | 33.33 ±3.19 (14.15) | 48.71 ±3.58 (16.75) | 89.84 ±1.48 (2.74) | 77.99 ±1.49 (0.23) | 77.87 ±1.32 | 2.96 ±0.87 |
| Retrained Baseline | 7.00 ±4.96 | 13.09 ±9.29 | 19.18 ±13.61 | 31.96 ±21.13 | 92.58 ±3.36 | 77.76 ±0.63 | 78.59 ±0.38 | 5.57 ±7.04 |

This table shows the accuracies of the unlearned model $\Phi_{G_u}$ on the unlearning dataset $D_u$, its subgroups $\{T_p\}$ split by memorization scores and test dataset $D_{test}$, along with the local fairness metric across different Non-IID scenarios.

Table F3 presents the results of our approach under varying degrees of Non-IID distributions using the CIFAR-10 dataset. A Dirichlet distribution is employed to simulate data heterogeneity, where the concentration parameter $\alpha$ controls the degree of Non-IIDness. Overall, our approach performs robustly across different Non-IID scenarios. In the case where $\alpha = 10$, there is significant overlap between the remaining dataset and the unlearning dataset. Under this setting, our method achieves performance comparable to the retrained baseline across all memorization score groups. Specifically, the accuracy of the entire unlearning dataset on the unlearned model using our method reaches 77.99%, closely matching the retrained baseline at 77.76%. Moreover, our method ensures fairness comparable to retrained baselines. However, the fine-tuning approach fails to unlearn any information, maintaining nearly 100% accuracy across all memorization score subgroups. This suggests that

fine-tuning alone is insufficient to remove significantly overlapping information. On the other hand, when $\alpha \rightarrow 0$, indicating no overlap between clients' data, fine-tuning alone can achieve optimal unlearning performance. This observation is consistent with the principle of catastrophic forgetting (Goodfellow et al., 2013), where fine-tuning only on the remaining dataset may naturally cause the model to forget information associated with the unlearning dataset if there is no information overlap between them.

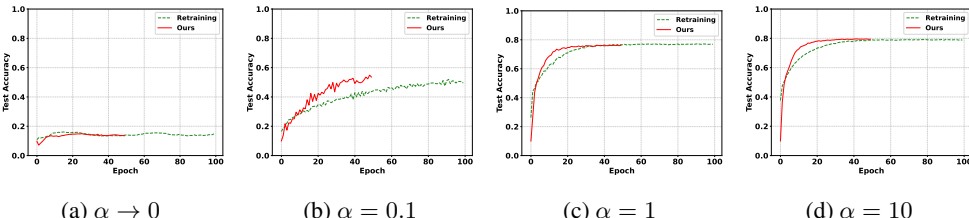

| (a) $\alpha \rightarrow 0$ | (b) $\alpha = 0.1$ | (c) $\alpha = 1$ | (d) $\alpha = 10$ |

Figure F2: Efficiency comparison between unlearning and retraining for different Non-IID scenarios.

Due to the unlearning failure of the only fine-tuning approach, we conduct the time analysis exclusively on our method. Figure F2 illustrates the comparison of test performance over time between our method and the retrained baseline. In cases where $\alpha \rightarrow 0$ and $\alpha = 0.1$, representing extreme Non-IID scenarios, the behavior is relatively unstable. For more general distributions, our method achieves approximately a 50% improvement in time efficiency compared to the retrained baseline. This demonstrates that our method is time-efficient across various distribution scenarios.

### APPENDIX F.3 IMPORTANT OR REDUNDANT PARAMETERS SELECTION

In this section, we compare different parameter selection methods. There are several existing machine unlearning methods based on the removal of important parameters, in contrast to our approach, which focuses on the removal of redundant parameters. Important parameters are defined as those that contribute significantly to learning, typically characterized by large gradients or large weights. These methods aim to remove important parameters associated with the unlearning dataset $D_u$ and require access to $D_u$ during the unlearning process. Although the unlearning dataset $D_u$ is not accessible during the unlearning stage in our setting, and these methods are originally designed as machine unlearning algorithms that may not be suitable for federated learning systems, we relax these requirements and adopt these parameter selection methods as comparative baselines. The methods we select included:

- **SalUn** (Fan et al., 2024). The **SalUn Selection** selects parameters with the highest gradient magnitudes, based on the principles of saliency maps.

- **Localized Strategy** (Torkzadehmahani et al., 2024). The **Localized Strategy Selection** is a recently proposed parameter selection method that chooses channel-wise neurons with the highest magnitude of the weighted gradient (i.e., the product of parameters and gradients).

- **Deep Layers**. The **Deep Layers Selection** is motivated by previous studies on the structural localization of unlearning information in neural networks (Maini et al., 2023). This method adopts the selection strategy of **SalUn** (Fan et al., 2024) but restricts the selection to deep layers.

- **Shallow Layers**. The **Shallow Layers Selection** follows the same selection method as **Deep Layers Selection**, but focuses on selecting parameters from shallow layers.

*(Noted. we only apply the parameter selection strategies of these methods, not their completed frameworks.)*

We conduct the experiment under the CIFAR-10 IID setting and attempt to unlearn one client out of ten clients. The results of the unlearning process are summarized in Table F4. These outcomes highlight the trade-offs between different parameter selection strategies and their effects on both generalization and unlearning performance across various memorization score subgroups.

Table F4: Performance comparison of FedMemEraser vs. baselines with different selection strategies

| Parameter Selection | Accuracy by Unlearning Memorization Score Subgroup (%) ($\Delta \downarrow$) | | | | | Unlearning Dataset Acc. (%) ($\Delta \downarrow$) | Test Dataset Acc. (%) ↑ | Local Fairness ($10^{-3}$) ↓ |
|---|---|---|---|---|---|---|---|---|
| | Group (95%,100%] | Group (90%,95%] | Group (85%,90%] | Group (80%,85%] | Group (0%,80%] | | | |
| SalUn | 15.20 ±1.18 (**2.40**) | 41.33 ±1.24 (**6.93**) | 60.00 ±0.57 (17.60) | 75.60 ±1.13 (16.53) | 95.15 ±0.92 (3.97) | 86.40 ±0.63 (3.91) | 70.29 ±0.27 | 0.35 ±0.06 |
| Localized | 23.60 ±3.71 (10.80) | 52.00 ±1.82 (17.60) | 74.13 ±1.54 (3.47) | 86.27 ±2.36 (5.86) | 98.40 ±0.12 (0.72) | 90.79 ±0.23 (**0.48**) | 74.40 ±0.30 | 0.32 ±0.03 |
| Deep | 26.80 ±0.57 (14.00) | 57.33 ±2.07 (22.93) | 78.53 ±0.38 (**0.93**) | 88.93 ±1.80 (**3.20**) | 98.79 ±0.17 (**0.33**) | 91.84 ±0.19 (1.53) | 74.65 ±0.23 | **0.13** ±0.02 |
| Shallow | 18.80 ±3.31 (6.00) | 44.00 ±1.42 (9.60) | 65.87 ±0.50 (11.73) | 78.00 ±3.31 (14.13) | 95.61 ±0.12 (3.51) | 87.48 ±0.22 (2.83) | 72.87 ±0.45 | 0.54 ±0.08 |
| Ours | 20.93 ±1.80 (8.13) | 49.73 ±1.68 (15.33) | 68.00 ±1.82 (9.60) | 84.67 ±1.24 (7.46) | 96.96 ±0.85 (2.16) | 89.36 ±0.67 (0.95) | 73.67 ±0.72 | 0.50 ±0.05 |
| Retrained Baseline | 12.80 ±9.34 | 34.40 ±19.80 | 77.60 ±14.71 | 92.13 ±5.58 | 99.12 ±0.62 | 90.31 ±0.16 | 74.79 ±0.38 | 0.01 ±0.00 |

This table demonstrates the accuracies of the unlearned model $\Phi_{G_u}$ on the unlearning dataset $D_u$, its subgroups $\{T_p\}$ split by memorization scores and test dataset $D_{test}$, along with the local fairness metric across different parameter selection methods.

Firstly, we observe that the results are relatively similar across different parameter selection strategies. Specifically, the **SalUn Selection** approach, which ranks parameters by saliency, results in significant drops in accuracy across all memorization score subgroups, particularly in the high-score ranges. A similar trend is observed in the **Shallow Layers Selection** strategy. This suggests that these methods tend to remove parameters that are crucial for representing the unlearning dataset $D_u$, often leading to excessive forgetting and degradation in performance on both the unlearning dataset and the test dataset. For the entire unlearning dataset, the accuracies drop to 86.40% and 87.48%, compared to 90.31% for the retrained model, indicating excessive forgetting. Moreover, the test accuracies decrease to 70.29% and 72.87%, respectively, in comparison to the baseline accuracy of 74.79%, implying reduced generalization performance. In contrast, the **Localized Strategy Selection** and **Deep Layers Selection** strategies preserve some memorization information, as evidenced by the higher accuracies in the top memorization score group: 23.60% and 26.80%, respectively, compared to the baseline of 12.80%.

These findings suggest that important parameters for the unlearning dataset mainly encode shared and general information. Removing these parameters may disable some memorization and shared representations, while shared information may be recovered through fine-tuning on the remaining dataset $D_r$. That is feasible, but our proposed method demonstrates greater precision. Unlike **SalUn Selection** and **Shallow Layers Selection**, which tend to forget excessive information, and unlike **Localized Strategy Selection** and **Deep Layers Selection**, which retain some memorization information, our method maintains a moderate and balanced effect across all memorization score subgroups. It specifically targets parameters associated with memorization. Further discussion on the relationship between parameters and information is provided in Appendix H.1. For local fairness, all selection strategies demonstrate comparable performance.

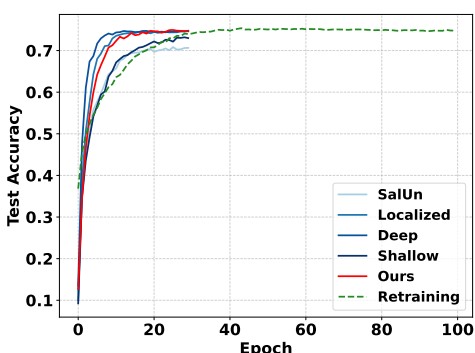

Figure F3: Efficiency comparison across different parameter selection methods

Regarding time efficiency, the observed trends are expected. As shown in Figure F3, our method also demonstrates moderate runtime performance. The **SalUn Selection** and **Shallow Layers Selection** strategies are more difficult to fine-tune due to excessive forgetting. Conversely, the **Localized Strategy Selection** and **Deep Layers Selection** approaches retain some memorization information, enabling quicker performance recovery.

In summary, our parameter selection method is both moderate and precise, making it particularly suitable for federated unlearning scenarios where the unlearning dataset $D_u$ is unavailable during the unlearning phase.

APPENDIX F.4 UNLEARNING MULTIPLE CLIENTS

In this subsection, we evaluate the effectiveness of our method in a multi-client unlearning setting. Specifically, we simulate the removal of 2, 4, 6, and 8 clients from a federated learning system comprising 10 clients under a CIFAR-10 Non-IID partitioning. Table F5 presents the accuracy of the unlearned model $\Phi_{G_u}$ across memorization-based subgroups of the unlearning dataset $D_u$, as well as the overall performance on $D_u$ and the general test set $D_{\text{test}}$. It also provides the evaluation of local fairness.

Table F5: Performance of FedMemEraser under multiple clients unlearning scenarios

| Client | Accuracy by Unlearning Memorization Score Subgroup (%) ($\Delta \downarrow$) | | | | | Unlearning Dataset Acc. (%) ($\Delta \downarrow$) | Test Dataset Acc. (%) $\uparrow$ | Local Fairness ($10^{-3}$) $\downarrow$ |
|---|---|---|---|---|---|---|---|---|
| | Group (95%,100%] | Group (90%,95%] | Group (85%,90%] | Group (80%,85%] | Group (0%,80%] | | | |
| **2 Clients ($\rho = 0.4$)** | | | | | | | | |
| Ours | 18.44 ±1.71 (6.56) | 25.06 ±0.95 (5.96) | 34.26 ±1.16 (12.14) | 40.68 ±0.95 (10.13) | 87.11 ±0.16 (3.06) | 76.89 ±0.17 (0.52) | 73.17 ±0.31 | 88.80 ±5.52 |
| Retrained Baseline | 11.88 ±8.44 | 19.10 ±13.53 | 22.12 ±15.51 | 30.55 ±13.82 | 90.17 ±4.13 | 77.41 ±0.77 | 73.90 ±0.69 | 68.54 ±14.64 |
| **4 Clients ($\rho = 0.4$)** | | | | | | | | |
| Ours | 19.99 ±1.61 (7.47) | 27.14 ±1.99 (9.32) | 30.17 ±1.57 (10.57) | 33.40 ±2.17 (9.79) | 76.22 ±1.69 (3.65) | 68.65 ±1.59 (1.22) | 68.79 ±1.09 | 87.88 ±4.86 |
| Retrained Baseline | 12.52 ±8.93 | 17.82 ±12.62 | 19.60 ±13.89 | 23.61 ±16.70 | 79.87 ±2.41 | 69.87 ±0.44 | 68.78 ±0.30 | 82.42 ±10.46 |
| **6 Clients ($\rho = 0.5$)** | | | | | | | | |
| Ours | 18.22 ±1.01 (5.95) | 22.25 ±1.43 (8.22) | 24.05 ±1.23 (8.45) | 27.11 ±1.79 (9.21) | 67.54 ±0.29 (1.49) | 62.65 ±0.31 (0.02) | 64.62 ±0.35 | 52.73 ±8.26 |
| Retrained Baseline | 12.27 ±8.69 | 14.03 ±9.93 | 15.60 ±11.03 | 17.90 ±12.70 | 69.03 ±2.79 | 62.63 ±0.50 | 64.06 ±0.28 | 70.37 ±17.79 |
| **8 Clients ($\rho = 0.6$)** | | | | | | | | |
| Ours | 13.10 ±1.06 (6.96) | 16.00 ±1.41 (6.52) | 18.76 ±1.11 (8.43) | 21.37 ±0.70 (9.00) | 49.21 ±0.92 (0.61) | 48.83 ±0.81 (0.66) | 51.87 ±0.57 | 417.23 ±17.96 |
| Retrained Baseline | 6.14 ±4.36 | 9.48 ±6.75 | 10.33 ±7.34 | 12.37 ±8.75 | 49.82 ±2.33 | 48.17 ±1.44 | 50.86 ±1.43 | 119.65 ±26.38 |

This table shows the accuracies of the unlearned model $\Phi_{G_u}$ on the unlearning dataset $D_u$, its subgroups $\{T_p\}$ split by memorization scores and test dataset $D_{test}$, along with the local fairness metric across multiple clients unlearning scenarios.

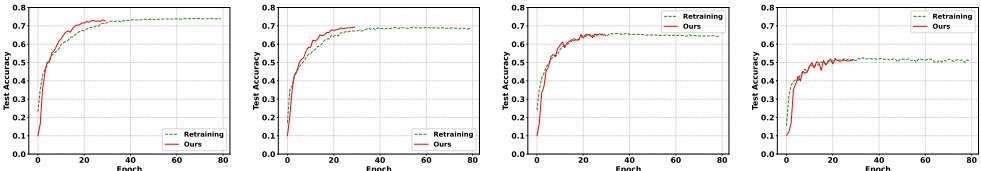

(a) Unlearning 2 Clients    (b) Unlearning 4 Clients    (c) Unlearning 6 Clients    (d) Unlearning 8 Clients

Figure F4: Efficiency comparison between unlearning and retraining for multiple clients.

The results demonstrate that our method consistently achieves effective unlearning across all configurations. Compared to the retrained baseline, our approach produces similar subgroup accuracies, while maintaining competitive or superior performance on the test dataset. For instance, when unlearning 8 clients, our method achieves a substantial reduction in high memorization group accuracies (e.g., 13.10% vs. 6.14% in the [95%, 100%] group), and still retains a test accuracy of 51.87%. This trend is consistent in all unlearning cases: our approach consistently achieves similar accuracies on memorization-based subgroups while maintaining meaningful accuracy on the test dataset. Furthermore, in local fairness evaluation, our approach maintains fairness levels that are largely consistent with the retrained baselines, except in the case of unlearning 8 out of 10 clients. In this extreme setting, removing too much learned knowledge may introduce uncontrolled randomness.

Considering time consumption, Figure F4 illustrates a clear efficiency advantage of our method over the retraining baselines when unlearning a small proportion of clients (i.e., fewer than 50%). However, when unlearning the majority of the knowledge from the federated learning system, the shared information learned from the entire dataset $D$ are disrupted. Consequently, essential patterns are removed, causing the model to degrade toward the random state. As a result, the fine-tuning process begins to resemble full retraining, particularly when unlearning 8 out of 10 clients. This phenomenon underscores an extreme scenario where removing common patterns, instead of specific memorization information, can disable the model and make the unlearning become retraining.

Table F6: Performance of FedMemEraser under various large-scale FL settings

| Client | Accuracy by Unlearning Memorization Score Subgroup (%) ($\Delta \downarrow$) | | | | | Unlearning Dataset Acc. (%) ($\Delta \downarrow$) | Test Dataset Acc. (%) $\uparrow$ | Local Fairness ($10^{-3}$) $\downarrow$ |
|---|---|---|---|---|---|---|---|---|
| | Group (95%,100%] | Group (90%,95%] | Group (85%,90%] | Group (80%,85%] | Group (0%,80%] | | | |
| 10 Clients | | | | | | | | |
| Ours | 20.27 ±1.54 (7.47) | 50.40 ±3.44 (16.00) | 70.93 ±2.29 (6.67) | 85.60 ±0.86 (6.53) | 98.66 ±0.20 (0.46) | 90.56 ±0.35 (0.25) | 74.97 ±0.27 | 0.91 ±0.06 |
| Retrained Baseline | 12.80 ±9.34 | 34.40 ±19.80 | 77.60 ±14.71 | 92.13 ±5.58 | 99.12 ±0.62 | 90.31 ±0.16 | 74.79 ±0.38 | 0.14 ±0.01 |
| 20 Clients | | | | | | | | |
| Ours | 41.87 ±1.36 (3.20) | 54.13 ±1.51 (6.13) | 64.80 ±0.65 (5.60) | 71.20 ±1.13 (2.13) | 95.01 ±0.11 (1.56) | 87.56 ±0.23 (1.25) | 73.56 ±0.18 | 5.22 ±0.08 |
| Retrained Baseline | 38.67 ±7.64 | 48.00 ±12.26 | 70.40 ±9.13 | 73.33 ±2.38 | 96.57 ±0.04 | 88.81 ±0.30 | 74.51 ±0.21 | 6.54 ±0.26 |
| 50 Clients | | | | | | | | |
| Ours | 58.00 ±3.27 (8.00) | 60.67 ±1.89 (1.34) | 62.67 ±0.94 (7.33) | 74.00 ±1.63 (1.33) | 91.00 ±0.28 (0.37) | 85.00 ±0.33 (0.40) | 75.61 ±0.13 | 7.78 ±0.65 |
| Retrained Baseline | 50.00 ±8.83 | 59.33 ±10.94 | 70.00 ±4.63 | 72.67 ±3.94 | 90.63 ±0.10 | 84.60 ±0.24 | 74.88 ±0.11 | 9.34 ±0.19 |

This table shows the accuracies of the unlearned model $\Phi_{G_u}$ on the unlearning dataset $D_u$, its subgroups $\{T_p\}$ split by memorization scores and test dataset $D_{test}$, along with the local fairness metric across different large-scale FL unlearning scenarios.

## APPENDIX F.5    EFFECT OF CLIENT NUMBER

In this subsection, we examine how large-scale federated learning (FL) settings impact our algorithm. In principle, our approach targets the removal of redundant parameters; therefore, increasing the number of clients in an FL configuration is not expected to significantly influence the performance of our unlearning algorithm.

Table F6 presents the performance of our unlearned model under various large-scale FL settings. Specifically, our method achieves performance gaps of 7.47%, 3.20%, and 8.00% on the top memorized subgroup compared to the retrained baselines for configurations with 10, 20, and 50 clients, respectively. Furthermore, the generalization and fairness of the unlearned models remain comparable to those of the retrained baselines across all settings. Figure F5 illustrates the time efficiency of our unlearning algorithm. Notably, FedMemEraser maintains its computational efficiency under different large-scale FL settings.

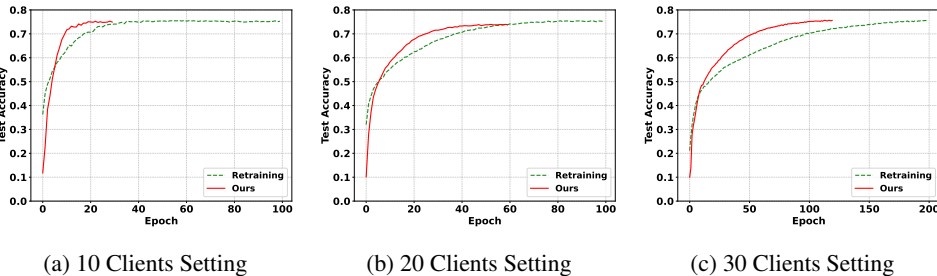

(a) 10 Clients Setting     (b) 20 Clients Setting     (c) 30 Clients Setting

Figure F5: Efficiency comparison between unlearning and retraining under various large-scale FL settings.

## APPENDIX G    REVISITING EXISTING FEDERATED UNLEARNING EVALUATION METRICS

In this section, we will revisit the unlearning evaluation metrics through the lens of memorization. Federated unlearning algorithms could be evaluated through multiple metrics. However, from the perspective of memorization, some of these metrics are unrealizable, certain metrics prove to be impractical, as they cannot definitively confirm that the influence of the unlearning data has been completely removed. Within the context of the unlearning problem, retraining is the gold standard, which basically means that all unlearned models are expected to approach the retrained model. Therefore, in this section, we mainly employ retrained models to analyze these metrics and initially investigate the tasks that clients identify as unlearning targets.

## APPENDIX G.1    REVISITING METRICS BASED ON PARAMETER DIFFERENCES

The parameter difference assesses the differences between the unlearned model and the retrained model within the parameter space. If the unlearned model becomes more similar to the retrained model, it indicates a successful unlearning result. The common parameter difference metrics include *L2 distance* (Shaik et al., 2024; Wang et al., 2023), *Kullback–Leibler divergence* (Gao et al., 2024; Wang et al., 2023) and *cosine similarity* (Liu et al., 2021; Lin et al., 2024).

However, these metrics do not consider the relationship between the unlearning data and the remaining data. If the unlearning dataset is closely similar to the remaining dataset, the difference in parameters between the unlearned model and the original model may become insignificant. Additionally, this metric does not account for the stochastic nature of retraining. Models retrained on the same remaining dataset can still reach different local optimal points. We conducted an easy experiment to demonstrate our findings.

**Experiment**. In this experiment, we mainly utilize Cifar-10 (Krizhevsky et al., 2009) as the dataset and Resnet18 (He et al., 2016) as the network architecture. Subsequently, we construct an FL framework involving 10 clients and simulate the process of unlearning one of them. Specifically, we begin by training an original model using all 10 clients, and to simulate the unlearning process, we exclude the client who requests unlearning and proceed to retrain several unlearned models with the remaining 9 clients. Here all unlearned models are retrained on the remaining clients. Finally, we will employ different parameter difference metrics on these models to support our viewpoint with evidence.

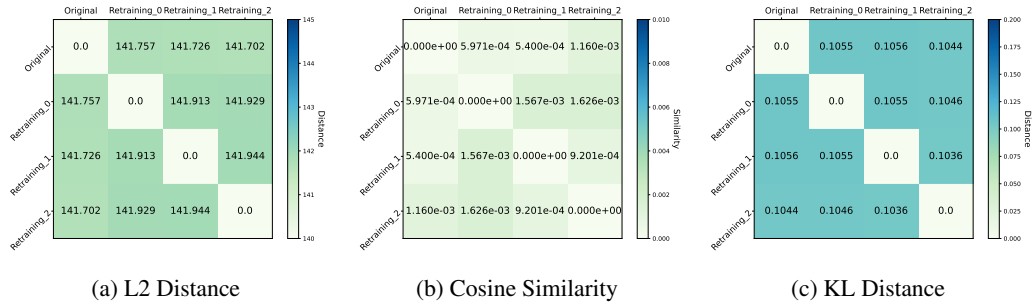

| (a) L2 Distance | (b) Cosine Similarity | (c) KL Distance |

Figure G6: Parameter differences measurement between models.

Table G7: Accuracies of original and retrained models

| Models | Training Dataset Acc.(%) | Test Dataset Acc.(%) | Unlearning Dataset Acc.(%) |
|---|---|---|---|
| Original Model | 95.01 | 76.08 | 99.37 |
| Retrained Model No.0 | 95.60 | 73.96 | 79.06 |
| Retrained Model No.1 | 95.81 | 74.52 | 80.28 |
| Retrained Model No.2 | 96.02 | 75.31 | 80.49 |

**Findings**. The performance of the models is presented in Table G7. In general, all models have successfully converged. In addition, retrained models show similar performance in the test dataset, achieving approximately 74% precision. Their performance in the unlearning dataset is also consistent, with accuracies around 80%.

In Figure G6, we illustrate three distinct distance metrics. The 'Original' refers to the model trained using the unlearning dataset, whereas the 'Retraining' refers to the models trained excluding the unlearning dataset. We create one original model and three retrained models, and then assess the distances among them. We have the following findings:

- **There exists the significant distance between the retrained models.** In Figure G6a, It has been noted that the L2 distances among the retrained models are all close to 141. Additionally, in Figure G6b, the cosine similarities between retrained models approaches 0. This suggests that these retrained models do not exhibit similarity in parameter space.

- **The distance between the retrained models is not significant compared to the original model.** For example, in Figure G6a, the L2 distance between the retrained model no.0 and the original model is $141.757$, however, the distance between the retrained model no.0 and no.1 is $141.913$, which is not significant. If apply the KL distance, the distances of retrained model no.0 to retrained model no.1 and the original model are both around $0.1055$, which shown in the Figure G6c. Therefore, when considering the retrained models, their distance proves to be relatively insignificant in comparison to the original model.

Basically, our experiments demonstrate that unlearned models may not approach the retrained models in the parameter space. In fact, even retrained models themselves can differ significantly from one another, highlighting the stochastic nature of training. While the unlearned model may successfully forget the unlearned dataset, it may still differ from the baseline retrained model. Additionally, if the unlearning dataset shares a similar feature distribution with the remaining data, this can lead to indistinguishability between the unlearned model and the original model in the parameter space. Overall, metrics based on parameter distance are not effective for evaluating unlearning.

### APPENDIX G.2  REVISITING METRICS BASED ON PERFORMANCE METRICS

Performance metrics like accuracy or loss of the unlearning dataset are commonly employed in the unlearning task to directly assess the unlearning results. Nonetheless, these metrics for the unlearning dataset overlook the distinctions between each unlearning example. The close accuracy or loss of the unlearning dataset, when compared to the retrained model performance, cannot guarantee the unlearning effect for each example. In the extreme case, some unlearning algorithms may primarily unlearn generalized examples in the unlearning dataset, resulting in similar accuracy compared to the retrained model, but this does not remove the impact of memorization information. Furthermore, retrained models might memorize examples randomly, resulting in a significant variation in how they perform on particular examples within the unlearning dataset. This makes it difficult for a single retrained model to serve as a reliable baseline. Our experiment would show our findings.

**Experiment**. In this experiment, we also utilize Cifar-10 (Krizhevsky et al., 2009) as the dataset and Resnet18 (He et al., 2016) as the network architecture. Subsequently, we still construct an FL framework involving 10 clients and simulate the process of unlearning one of them. Here, we directly retrain multiple global models with the remaining 9 clients. Then we can compare the example-level performance with these retrained models.

**Findings**. We evaluate the classification probability of each example within the unlearning dataset in multiple retrained models. We illustrate the results in Figure G7. It shows the correct classification probability of each example in the unlearning dataset on the different retrained models and these examples are sorted by the average correct classification probability. We can find:

- **Majority of examples are classified absolutely correctly.** Obviously, a substantial portion of the examples (examples with indices starting at 2000) are classified correctly even if these examples are not in the remaining dataset. This suggests that there is an overlap between the unlearning dataset and the remaining dataset. It indicates that the shared information will not be forgotten even using retraining.

- **A portion of examples have unstable performance.** Notably, there exists a portion of examples with varying classification probabilities on different retrained models. This variability implies that some examples are more sensitive to the stochastic retraining process.

Based on these findings, traditional performance metrics, such as accuracy or loss in the unlearning dataset, can be misleading. They fail to capture example-level differences that are critical to understanding whether individual data points have been effectively forgotten. Additionally, we notice that some examples exhibits unstable performance. This variability in classification probabilities is driven by rare or under-represented features in the remaining dataset (Feldman, 2020). During retraining, models may randomly memorize the information, resulting in inconsistent performance on these unstable examples. Therefore, performance metrics are not enough to measure unlearning, especially at the example-level.

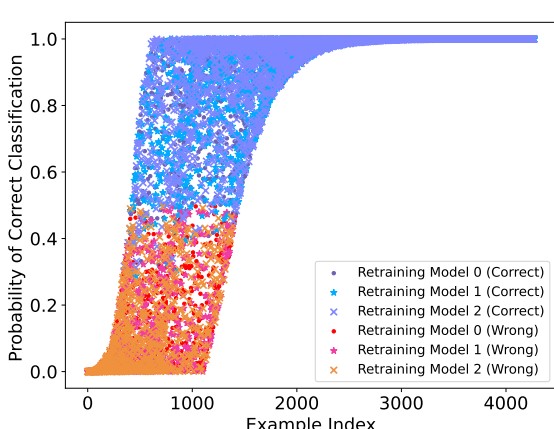

Figure G7: Classification probability of unlearning examples in retrained models.

### APPENDIX G.3    REVISITING BACKDOOR EVALUATION

In the context of unlearning, a backdoor evaluation is a typical assessment used to explicitly show the outcomes of the unlearning effect. In backdoor attacks, a backdoor trigger is injected into a portion of training data, and those are re-labeled to a pre-defined class. After the normal training, the original model will learn backdoor features. Therefore, in the subsequent unlearning step, if the backdoor features have been completely unlearned, the unlearning algorithm can be considered successful. Currently, this evaluation has widely used in many works (Zhao et al., 2023; Pan et al., 2025; Halimi et al., 2022; Li et al., 2023).

Nevertheless, it is observed that a fundamental issue in backdoor evaluation is that backdoor features are typically irrelevant to the unlearning dataset. In our Definition 2, when utilizing backdoor evaluation, it suggests that the backdoor features are purely memorization information and do not overlap with the remaining information. An unlearning algorithm being able to unlearn the unlearning dataset consisting of all backdoor features does not imply that it can also unlearn a dataset that includes intersections with the remaining information. We also conducted an experiment to support our opinion.

**Experiment**. In this experiment, we still utilize Cifar-10 (Krizhevsky et al., 2009) as the dataset and Resnet18 (He et al., 2016) as the network architecture. Subsequently, we still construct an FL framework involving 10 clients and simulate the process of unlearning one of them. To conduct the backdoor evaluation, we train two original models: one with the standard unlearning dataset and the other with the backdoor unlearning dataset. After this, we proceed by fine-tuning the two original models with the remaining clients to mimic the unlearning process and then evaluating the fine-tuned models' performance on the unlearning datasets.

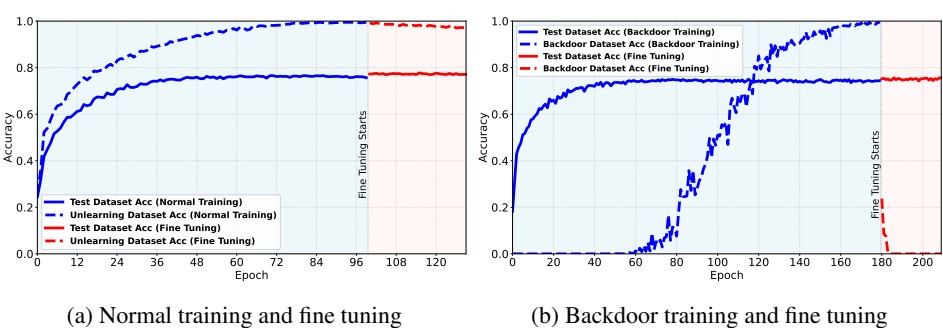

(a) Normal training and fine tuning      (b) Backdoor training and fine tuning

Figure G8: Backdoor vs normal training performance comparison in the fine-tuning stage.

**Findings**. We compare the performance of backdoor training and normal training in Figure G8. The figure G8a shows the trend in test accuracy (solid line) and accuracy on the unlearning dataset (dotted line) at the normal training stage with the unlearning dataset (blue background) and the fine-tuning stage without the unlearning dataset (red background). The figure G8b is similar, but the unlearning dataset is the backdoor dataset. Comparing the two sub-figures, we can observe:

- **For normal training, fine-tuning may not remove the influence of the unlearning dataset.** A slight decrease in the accuracy of the unlearning dataset is observed at the fine-tuning stage.
- **Fine-tuning on the remaining dataset can easily unlearn the backdoor dataset.** Backdoor dataset will be forgot quickly in the fine-tuning stage.

Obviously, backdoor evaluation overlooks the relationship between the unlearning dataset and the remaining dataset. The unlearning dataset may share general information with the remaining data, making it challenging to achieve effective unlearning through fine-tuning. In contrast, backdoor datasets typically contain conflicting information relative to the remaining dataset, which can facilitate unlearning during the fine-tuning process. Therefore, in the evaluation of unlearning algorithms, successful unlearning of a backdoor dataset does not necessarily indicate the overall effectiveness of the algorithm.

## APPENDIX H    DISCUSSION

### APPENDIX H.1    PARAMETERS AND INFORMATION REPRESENTATION

In this subsection, we investigate the relationship between model parameters and learned information.

Currently, the interpretability of neural networks and the way parameters encode information remain open research questions. Consequently, our analysis primarily provides a qualitative investigation of parameter resetting in the context of unlearning memorization information from the unlearning dataset.

According to prior interpretability studies based on saliency maps (Simonyan et al., 2014) and activations (Selvaraju et al., 2017), parameters exhibiting large gradients or large weighted gradients are typically considered important, as they are likely to encode information that mainly contributes to generalization performance. In contrast, other parameters are often unimportant or redundant and may correspond to memorization information. In the context of federated unlearning, we consider two main parameter selection strategies as discussed earlier: 1) resetting important parameters with respect to the unlearning dataset; and 2) resetting redundant parameters with respect to the remaining dataset.

Table H8: Unlearning performance comparison across different reset strategy

| Memorization Score Subgroups | Accuracy by Selection Strategy (%) | |
|---|---|---|
| | Important Parameters | Redundant Parameters |
| Group (90%,100%] | 19.00 | 29.60 |
| Group (80%,90%] | 24.60 | 42.00 |
| Group (70%,80%] | 23.60 | 42.20 |
| Group (60%,70%] | 24.80 | 49.60 |
| Group (0%,60%] | 28.52 | 52.37 |

This table presents the accuracies of the unlearned model $\Phi_{G_u}$ on the memorization subgroups $\{T_{p'}\}$ of the unlearning dataset $D_u$, with different parameter selection strategies applied during unlearning under the CIFAR-10 IID condition.

For the first strategy, resetting the important parameters associated with the unlearning dataset followed by fine-tuning on the remaining dataset can achieve unlearning, as shown in Appendix F.3. This works because the fine-tuning stage helps the model to relearn shared information while naturally forgetting some memorization information. This forgetting phenomenon resembles catas-

trophic forgetting Goodfellow et al. (2013), as it results from the domain shift between the unlearning dataset and the remaining dataset. As a result, in this fine-tuning stage, the model can forget some previous learned particular or memorization information. However, if the important parameters are not reset, fine-tuning alone may induce only minimal forgetting, and the process of unlearning might not succeed because gradient updates mainly maintain generalization performance.

The second strategy focuses on resetting redundant parameters, which is often more appropriate. As discussed in Appendix F.3, the first strategy might inadvertently retain some memorization information or erase too many general or shared patterns, since unlearning is primarily driven by natural forgetting during fine-tuning. In contrast, the second strategy explicitly targets redundant parameters that encode memorization information, offering a more precise unlearning mechanism. Fine-tuning subsequently enhances the forgetting effect further.

To empirically demonstrate the impact of parameter removal on different information types without fine-tuning, we conducted a simple experiment under the CIFAR-10 IID setting. Specifically, we reset approximately 5% of the model parameters using both selection strategies. Table H8 compares the resulting accuracies across various memorization subgroups of the unlearning dataset. We observe that resetting important parameters degrades accuracy across all memorization subgroups, with scores of 19.00%, 24.60%, 23.60%, 24.80%, and 28.52%, respectively. In contrast, resetting redundant parameters significantly reduces performance on the most memorized subgroup (to 29.60%) while preserving performance on less memorized subgroups (e.g., up to 52.37%). These results support the hypothesis that important parameters primarily encode shared information, whereas redundant parameters are more closely tied to memorization information. At the example level, memorization examples rely on memorization information and shared information, whereas generalized examples in the unlearning dataset primarily depend on shared information. Therefore, in the context of unlearning, resetting redundant parameters is more effective and preferable to resetting important ones.

### APPENDIX H.2  FAST MEMORIZATION PROXY

In Grouped Memorization Evaluation, we utilize memorization score (Feldman, 2020) to evaluate the memorization information in the examples. However, a major drawback of the memorization score is its high computational cost. To address this issue, a more effective approach is to apply quick memorization proxies linked to memorization scores. The most recent approach for estimating memorization is **Cumulative Sample Loss (CSL)** (Ravikumar et al., 2025).

CSL tracks how much loss a model accumulates on each sample over the entire training process. Specifically, CSL can be formulated as:

$$CSL((x_i, y_i)) = \sum_{t=0}^{T_{max}} \mathcal{L}(\Phi_G, (x_i, y_i)), \tag{15}$$

where $(x_i, y_i)$ denotes an input-label pair, $t$ represents the training iteration, and $\mathcal{L}$ is the loss function.

CSL is cost-free because it can be obtained directly from the standard training process without requiring any additional computation. During training, the loss for every sample is naturally computed at each iteration as part of the forward pass; CSL simply accumulates these existing per-sample losses over time.

Intuitively, easy samples receive low loss early and thus have low CSL, while hard or mislabeled samples accumulate much higher loss. CSL has been shown to closely match formal memorization measures but requires almost no additional computation. We conduct an experiment on CIFAR-10 to evaluate the alignment between the memorization score and CSL under the IID setting. A scatter plot comparison is provided in Figure H9, which demonstrates a positive correlation between CSL and the memorization score. Furthermore, we compute the cosine similarity between the two metrics, obtaining a value of 0.82. When identifying the top 20% most memorized examples, the two measures exhibit an overlap rate of 82%.

Therefore, when computational resources are limited, CSL serves as an effective alternative to the memorization score.

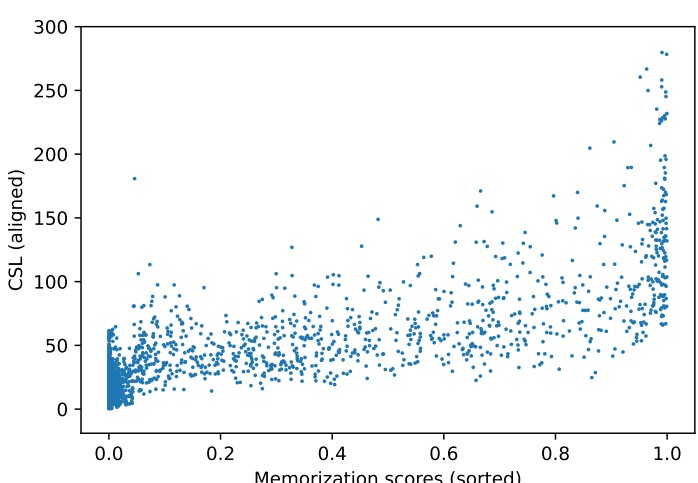

Figure H9: Relationship between CSL and memorization scores

## APPENDIX H.3   PRUNING RATIO SELECTION

In Appendix F.1, we demonstrate how the pruning ratio $\rho$ influences the unlearning performance. Therefore, in this section, we attempt to give a pruning ratio $\rho$ selection rule.

As we discussed before, the optimal value of $\rho$ depends on the distribution of memorization information across the model parameters. In other words, $\rho$ depends the proportion of redundant parameters of $D_r$. This is because the non-overlapping information of $D_u$ is irrelevant to the important parameters of $D_r$. Therefore, the main challenge lies in determining an appropriate threshold to identify important parameters of $D_r$. Subsequently, the pruning ratio $\rho$ can be set based on the proportion of redundant parameters.

Actually, various indicators (Fan et al., 2024; Maini et al., 2023) have been proposed to assess parameter importance for a given dataset on a neural network. For example, gradient-based input saliency maps (Adebayo et al., 2018) is a common tool. We apply this tool to analyze the parameter importance distribution on all unlearning tasks and provide the parameter importance distribution figures in Figure H10. Considering that the magnitudes of gradient updates often span multiple orders of magnitude, we apply the logarithmic function to visualize the distribution.

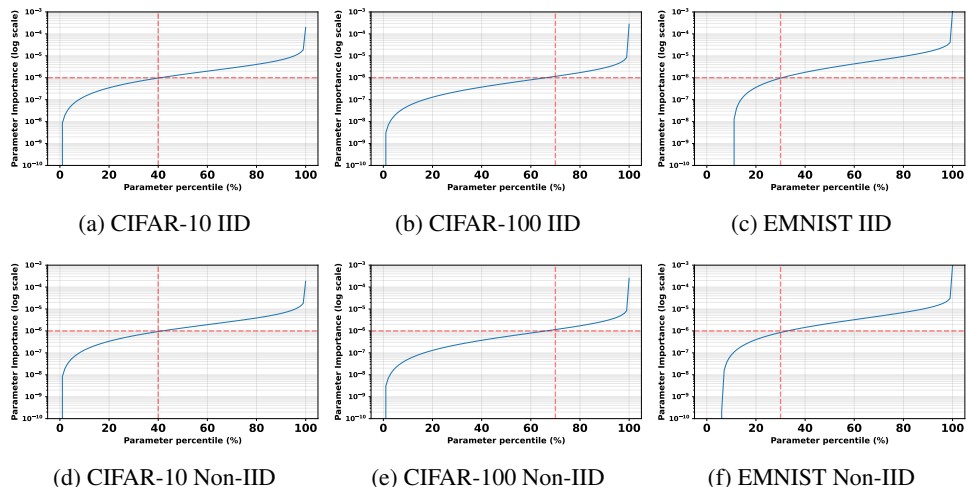

Figure H10: Parameter importance distribution.

We measure the parameter importance distributions on CIFAR-10, CIFAR-100, and EMNIST under both IID and Non-IID conditions. Figure H10 illustrates similar trends across them. Most parameters exhibit extremely small importance (on the order of $10^{-8}$ to $10^{-6}$), suggesting that they contribute minimally to the current optimization dynamics. As the percentile increases, parameter importance rises gradually and then sharply increases within the top few percent of parameters. This sharp rise typically begins around values of $10^{-5}$. This long-tailed distribution indicates that a small subset of parameters is very important, while the majority remain relatively inactive.

Based on this observation, we define parameters with importance values around or above the order of $10^{-5}$ as important. After that, we adaptively select the next lower order of magnitude, i.e., parameters with importance below the order of $10^{-6}$, as redundant parameters. Using this adaptive method, the resulting pruning ratios for CIFAR-10 and EMNIST are approximately 0.3 to 0.4, while for CIFAR-100, the pruning ratio reaches around 0.7. These values are close to our empirical selections.

Additionally, when the task involves unlearning a larger number of clients, the pruning ratio should be increased, as the amount of relevant non-overlapping information grows. Data augmentation also requires a higher pruning ratio, since it promotes feature generalization and tends to enlarge redundant parameters. From the perspective of unlearning effectiveness, it is therefore often preferable to select a larger pruning ratio.

### APPENDIX H.4    EXTENDING TO CLASS-LEVEL AND EXAMPLE-LEVEL UNLEARNING

**Class-level Unlearning**. Our approach can be easily applied to class-level unlearning without any modification. Moreover, class-level unlearning is generally easier than client-level unlearning since the overlapping information between different classes is minimal, which means we can apply a small pruning ratio $\rho = 0.2$.

We conduct class-level unlearning experiments with 10 clients on the CIFAR-10 dataset under both IID and Non-IID settings. The task involves unlearning all examples labeled as class 0 across all clients. Using FedMemEraser, the unlearning results are reported in Table H9, and a comparison of time consumption is presented in Figure H11.

Table H9: Class-level task unlearning performance comparison.

|  | Unlearning Dataset Acc. (%) | Test Dataset Acc. (%) |
| --- | --- | --- |
| CIFAR-10 IID |  |  |
| Ours | 0.00 | 68.45 |
| Retrained Baseline | 0.00 | 67.52 |
| CIFAR-10 Non-IID |  |  |
| Ours | 0.00 | 71.32 |
| Retrained Baseline | 0.00 | 68.44 |

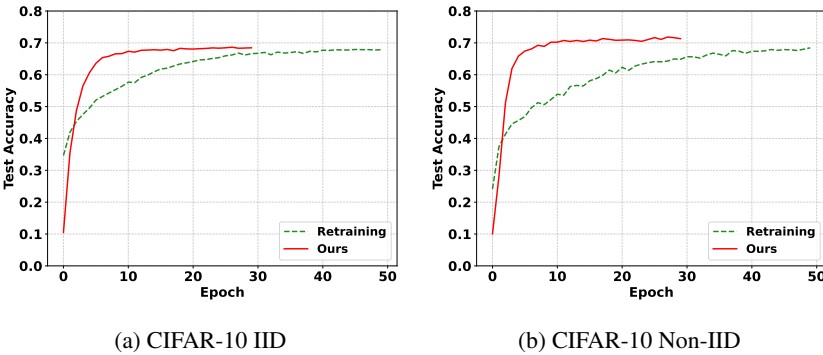

(a) CIFAR-10 IID                    (b) CIFAR-10 Non-IID

Figure H11: Efficiency comparison for the class-level unlearning task.

From the results, we observe that our method achieves complete unlearning performance on class 0 while maintaining strong generalization capability. Specifically, our method achieves unlearning

accuracies of 0% under the two conditions. In terms of generalization performance, the test accuracies of 68.45% and 71.32% under IID and Non-IID settings, respectively, outperform the retrained baselines, which achieve 67.52% and 68.44% accuracy. Additionally, in terms of time efficiency, our method is 3× faster than the retrained baselines, demonstrating a significant advantage.

**Example-level Unlearning**. Regarding example-level unlearning, FedMemEraser can be directly applied. For efficiency, the system may accumulate multiple single-example unlearning requests and store them in a request stack. Once the stack reaches capacity or satisfies another predefined condition, the system executes the unlearning algorithm to remove the knowledge of these examples. Following the pipeline of FedMemEraser, we continue to reset redundant parameters with respect to the $D_r$ and fine-tune on the $D_r$. Overall, in our approach, example-level unlearning is essentially similar to client-level unlearning.

We consider a scenario in which clients request the removal of 5,000 examples across 10 clients under both IID and non-IID settings using the CIFAR-10 dataset. Table H10 reports our results for this example-level unlearning scenario. Overall, the unlearning performance is comparable to client-level unlearning, achieving accuracy gaps of 5.33% and 1.00% on the most memorized subgroup. We also provide a time comparison in Figure H12, which illustrates the runtime difference between unlearning and full retraining. For both IID and non-IID cases, our approach consistently maintains a 50% improvement in efficiency.

Table H10: Example-level task unlearning performance comparison

| | Accuracy by Unlearning Memorization Score Subgroup (%) $(\Delta \downarrow)$ | | | | | Unlearning Dataset Acc. (%) $(\Delta \downarrow)$ | Test Dataset Acc. (%) ↑ | Local Fairness $(10^{-3}) \downarrow$ |
|---|---|---|---|---|---|---|---|---|
| | Group (95%,100%] | Group (90%,95%] | Group (85%,90%] | Group (80%,85%] | Group (0%,80%] | | | |
| CIFAR-10 IID | | | | | | | | |
| Ours | 24.00 ±1.18 (5.33) | 48.67 ±1.68 (5.20) | 65.47 ±4.35 (1.34) | 76.13 ±5.07 (8.14) | 96.44 ±0.90 (2.38) | 88.37 ±1.08 (1.28) | 73.83 ±1.04 | 4.06 ±0.76 |
| Retrained Baseline | 18.67 ±0.75 | 43.47 ±0.50 | 64.13 ±0.19 | 84.27 ±0.19 | 98.82 ±0.07 | 89.65 ±0.06 | 73.84 ±0.30 | 0.11 ±0.03 |
| CIFAR-10 Non-IID | | | | | | | | |
| Ours | 22.40 ±2.14 (1.00) | 33.20 ±1.18 (4.00) | 34.27 ±0.83 (2.81) | 39.73 ±2.17 (3.20) | 80.77 ±0.24 (0.29) | 74.20 ±0.23 (0.37) | 73.45 ±0.45 | 54.53 ±8.12 |
| Retrained Baseline | 21.40 ±0.20 | 29.20 ±0.57 | 31.46 ±0.50 | 42.93 ±1.86 | 81.06 ±0.62 | 74.57 ±0.27 | 74.89 ±0.14 | 25.86 ±15.54 |

This table shows the accuracies of the unlearned model $\Phi_{G_u}$ on the unlearning dataset $D_u$, its subgroups $\{T_p\}$ split by memorization scores and test dataset $D_{test}$, along with the local fairness metric under IID and Non-IID example-level unlearning.

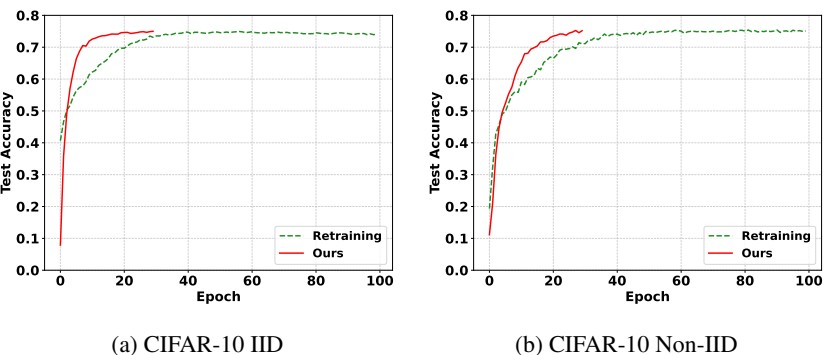

(a) CIFAR-10 IID          (b) CIFAR-10 Non-IID

Figure H12: Efficiency comparison for the example-level unlearning task.

### APPENDIX H.5 LOSS SURFACE OF THE SHORTEST UNLEARNING PATH

In this section, we investigate the loss surface along the shortest path from the original model to any retrained model. Specifically, we compute the model difference vector and use it as a direction with a fixed step size to update the original model parameters. This simulates the unlearning process along the shortest path, allowing us to observe changes in the loss surface.

Figure H13 illustrates the loss surface during the unlearning process under CIFAR-10 Non-IID condition. Notably, the loss increases sharply when deviating from the original model parameters and gradually decreases as the model approaches the retrained state. This observation indicates that the

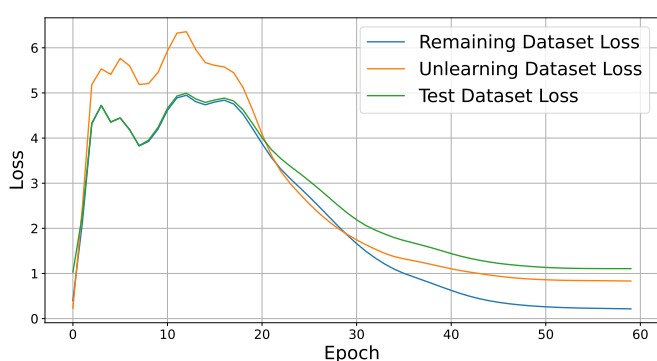

Figure H13: Loss surface of the shortest unlearning path

original and retrained models lie in different loss basins. Furthermore, even slight perturbations to the original model can substantially degrade performance. In the context of unlearning, where specific knowledge must be removed from the model, this suggests a high risk of impairing generalization. After unlearning, the model must re-fit to the remaining data distribution.

From the perspective of the shortest path, unlearning can be interpreted as a partial re-training process, rather than simply reverting to a previous state of the original model. In cases involving substantial data removal, post-training or fine-tuning may be necessary to recover generalization performance. Moreover, our unlearning process exhibits behavior similar to that of the shortest path, and the observed transformation of the loss surface supports the validity of our unlearning algorithm.

