# OpenReview forum: "Understanding Federated Unlearning through the Lens of Memorization"
_ICLR.cc/2026/Conference — Submitted to ICLR 2026_

### Official Review · Reviewer_hFAH · 2025-10-28

**Soundness:** 2
**Presentation:** 3
**Contribution:** 2
**Rating:** 4
**Confidence:** 3

**Summary:**

The manuscript revisits the problem of federated unlearning from a memorization perspective, arguing that only the unique memorized information within the forgetting dataset should be removed, while the shared patterns should be retained. The authors propose a metric for distinguishing between memorized and shared knowledge at the instance level: Grouped Memorization Evaluation, and introduce Federated Memorization Eraser, with experiments validating the effectiveness of their method. However, there remain several issues that merit further attention:

**Strengths:**

1. The manuscript redefines the federated unlearning problem from the perspective of memorization and demonstrates that overlapping or shared information should not be unlearned.
2. The manuscript proposes Grouped Memorization Evaluation, a novel metric that can measure memorization information at the example level, thereby enabling a fine-grained assessment of unlearning efficacy.

**Weaknesses:**

1. The effectiveness of the FedMemEraser method relies on the pruning ratio. The manuscript notes that this ratio needs to be determined empirically, which might increase the difficulty and tuning cost of applying the method in different scenarios.
2. The method's core assumption is that redundant parameters relative to the remaining dataset D_r  primarily carry the unique memorization information of the unlearning dataset D_u. The universality and completeness of this link require further theoretical or experimental validation.
3. The proposed "Grouped Memorization Evaluation" metric requires retraining the model multiple times to compute the memorization score for each example. This could introduce significant computational overhead in practice.
4. The results in Fig. 1 are too densely presented, which negatively impacts readability.

**Questions:**

1. In Section 6.2, during Stage 1, the manuscript uses the average gradient magnitude to evaluate the amount of memorized information in the parameters. However, the authors do not provide sufficient justification and evidence for the superiority and necessity of this step. Moreover, the rationale behind the selection of the threshold hyperparameter for filtering is not thoroughly explained—how is the predetermined percentage of initialized parameters determined? In addition, the method does not appear to account for sensitivity differences across layers.
2. The manuscript discusses the issue of overlapping and non-overlapping information. Since the proposed method aims to remove non-overlapping information while preserving overlapping information, would the presence of retained clients whose data distributions are similar to that of the forgetting client affect the evaluation results?
3. The core assumption that “memorized information is equivalent to non-overlapping information” appears to be overly idealized. In Equation (3), $F_m=F_u-F_g$ is defined as the memorized information, but the manuscript does not explain how to concretely distinguish between “overlapping” and “non-overlapping” components in the feature space. Given that the feature distributions are highly nonlinear, an empirical definition alone cannot guarantee that $F_m$ truly corresponds to the memorized portion.
4. The assumption regarding redundant parameters in FedMemEraser lacks validation in Stage 1. The key premise of the algorithm is that small-gradient parameters correspond to memorized information. This assumption is too absolute and lacks theoretical justification or empirical support.
5. FedMemEraser consists of three phases: positioning, resetting, and fine-tuning. If only positioning and reset phases are performed without fine-tuning, what would happen to the forgetting effect and generalization performance of the model? How much forgetting effect does the reset operation itself contribute, and how much does the subsequent fine-tuning contribute?
6. The manuscript's experiments mainly focus on client level forgetting. How will the FedMemEraser method apply to sample-level or category-level forgetting in federated learning?

---

> ### Author Response · Authors · 2025-11-24
> **Feedback for Reviewer hFAH - Part 1**
>
> Thank you for your insightful comments. We are glad to address your concerns.
>
> **Response to Weakness 1:**
>
> The pruning ratio $\rho$ is a key hyperparameter in our method, and we revisit the strategy for selecting $\rho$. The primary factor influencing $\rho$ is the distribution of non-overlapping information within the parameter space. Following the fundamental principle that the non-overlapping information of $D_u$ is irrelevant to the important parameters of $D_r$, the main challenge lies in determining an appropriate threshold to identify important parameters of $D_r$. Subsequently, the pruning ratio $\rho$ can be set based on the proportion of important parameters.
>
> Here, we propose an adaptive method to select $\rho$ based on the distribution of parameter importance. Various indicators can be used to assess parameter importance for a given dataset. For example, gradient-based input saliency maps [1] are a common tool. We analyse the parameter importance distribution on all unlearning tasks and provide the parameter importance distribution figures in **Appendix H.3**. Considering that the magnitudes of gradient updates often span multiple orders of magnitude, we apply the logarithmic function to visualize the distribution.
>
> We measure the parameter importance distributions on three datasets under both IID and Non-IID conditions, and observe similar trends across them. Most parameters exhibit extremely small importance (on the order of $10^{-8}$ to $10^{-6}$), suggesting that they contribute minimally to the current optimization dynamics. As the percentile increases, parameter importance rises gradually and then sharply increases within the top few percent of parameters. This sharp rise typically begins around values of $10^{-5}$. This long-tailed distribution indicates that a small subset of parameters is very important, while the majority remain relatively inactive. Based on this observation, we define parameters with importance values around or above the order of $10^{-5}$ as important. After that, we adaptively select the next lower order of magnitude, i.e., parameters with importance below the order of $10^{-6}$, as redundant parameters. Using this adaptive method, the resulting pruning ratios for CIFAR-10 and EMNIST are approximately 0.4, while for CIFAR-100, the pruning ratio reaches around 0.7. These values are close to our empirical selections.
>
> Additionally, when the task involves unlearning a larger number of clients, the pruning ratio should be increased, as the amount of relevant non-overlapping information grows. Data augmentation also requires a higher pruning ratio, since it promotes feature generalization and tends to enlarge redundant parameters. From the perspective of unlearning effectiveness, it is therefore often preferable to select a larger pruning ratio.

---

> ### Author Response · Authors · 2025-11-24
> **Feedback for Reviewer hFAH - Part 2**
>
> **Response to Weakness 2:**
>
> For the core assumption, we can provide a brief theoretical analysis and experimental validation.
>
> Considering relevant interpretability studies in neural networks [7], parameters with large gradients tend to learn more information about the corresponding data. Therefore, important parameters with respect to the remaining dataset $D_r$ primarily encode the information contained in $D_r$. Moreover, it is known that the unique memorization information of $D_u$ cannot be provided by $D_r$. Consequently, we can naturally infer that the unique memorization information of the unlearning dataset $D_u$ is more closely associated with the redundant parameters relative to the remaining dataset $D_r$.
>
> Specifically, we can define the information in $D_r$ as $F_r$, including both shared patterns and dataset-specific memorization. Similarly, we define $F_m$ to capture all information contained in $D_u$. Given that $D_u$ and $D_r$ may share patterns, the relevant parameters are expected to align during training [6]. Therefore, there exists an intersection $F_g = F_m \cap F_r$, which denotes the shared or overlapping information between the two datasets.  Furthermore, the unique memorization information or non-overlapping information of $D_u$ could $F_u^{mem} = F_u - F_g$ (In Section 4.1 $F_m = F_u^{mem}$). Similarly, the memorization information unique to $D_r$ are given by  $F_r^{mem} = F_r - F_g$. Since memorization information is assumed to be unique to each dataset, we have the disjointness condition: $F_r^{mem} \cap F_u^{mem} = \emptyset$. Our goal is to unlearn the unique memorization information in $D_u$, which requires locating $F_r^{mem}$. However, we do not have access to $D_u$. We can know $F_r \cap F_u^{mem} = \emptyset$. This implies that the important parameters of $D_r$ are unrelated to the unique memorization (i.e., non-overlapping) information in $D_u$. Consequently, the unique memorization information of the unlearning dataset $D_u$ must be embedded in the redundant parameters with respect to the remaining dataset $D_r$.
>
> For experimental validation, we provide a detailed discussion in **Appendix H.1**. We consider two primary parameter selection strategies: (1) resetting important parameters with respect to the unlearning dataset, and (2) resetting redundant parameters with respect to the remaining dataset. We then evaluate the performance across memorization-based subgroups.  The results indicate that resetting important parameters with respect to the unlearning dataset leads to a uniform performance degradation across all memorization-based subgroups. set. We observe that resetting important parameters degrades accuracy across all memorization subgroups, with scores of 19.00%, 24.60%, 23.60%, 24.80%, and 28.52%, respectively. In contrast, resetting redundant parameters significantly reduces performance on the most memorized subgroup (to 29.60%), while preserving performance on less memorized subgroups (e.g., up to 52.37%).
>
> To summarize, these results suggest that the redundant parameters with respect to the remaining dataset $D_r$ may encode the unique memorization information of the unlearning dataset $D_u$, which is a reasonable assumption. Table 1 further supports this perspective by demonstrating successful unlearning outcomes under this assumption.

---

> ### Author Response · Authors · 2025-11-24
> **Feedback for Reviewer hFAH - Part 3**
>
> **Response to Weakness 3:**
>
> Grouped Memorization Evaluation (GME) is a fine-grained evaluation framework designed to assess the effectiveness of unlearning. The primary computational cost arises from calculating the memorization scores, and this heavy spending has been criticized for a long time.  In this work, we employ memorization scores to adhere to the standard and rigorous memorization evaluation protocol. However, if the computational resources are limited, there exist fast and cost-free memorization proxies that can serve as approximations of the memorization score. The typical proxy is cumulative sample loss (CSL) [2], which tracks how much loss a model accumulates on each sample over the entire training process. Intuitively, easy samples receive low loss early and thus have low CSL, while hard or mislabeled samples accumulate much higher loss.
>
> CSL is cost-free because it can be obtained directly from the standard training process without requiring any additional computation. During training, the loss for every sample is naturally computed at each iteration as part of the forward pass; CSL simply accumulates these existing per-sample losses over time. Additionally, CSL has been shown to closely match formal memorization measures but requires almost no additional computation. We conduct an experiment and demonstrate that CSL could approximate the memorization scores. Specifically, the memorization score and CSL exhibit a 0.82 cosine similarity across the set $D_u$ under the CIFAR-10 IID condition. For detecting the top 20% most memorized examples, the two metrics show an overlap rate of 82%. We have added a new section in the revised version to discuss this solution in **Appendix H.2**.
>
>
>
> **Response to Weakness 4:**
>
> Thank you for your comment. We have revised Figure 1 in the manuscript and reduced data points to improve the readability.

---

> ### Author Response · Authors · 2025-11-24
> **Feedback for Reviewer hFAH - Part 4**
>
> **Response to Question 1:**
>
> According to interpretability studies [7] and memorization localization methods [4], evaluating gradient magnitude is a commonly used technique for mainly assessing parameter importance. In our approach, we can utilize small gradients, computed with respect to the remaining dataset, to identify redundant parameters effectively. The average gradient magnitude represents the gradient of $D_r$, which corresponds to the aggregated gradients from all remaining clients. The $D_r$ denotes the union of all local datasets belonging to the remaining clients.
>
> Another reason for using gradient magnitude is that it reflects the variance of the loss with respect to the model parameters. A small gradient magnitude indicates that variations in the corresponding parameter have little influence on the loss for the respective data. Therefore, redundant parameters with small gradient magnitudes have minimal impact on the loss $D_r$. Given that the non-overlapping information in $D_u$ cannot be provided by $D_r$, small gradient magnitudes may effectively identify parameters that are irrelevant to $D_r$ and potentially carry non-overlapping information from $D_u$.
>
> For details on pruning ratio selection, please refer to the **Response to Weakness 1**.
>
> Based on findings from memorization localization studies [4], memorization-related neurons are distributed across all layers of the network. Therefore, we apply the small-gradient criterion across all layers to detect redundant parameters and find that the gradient magnitude naturally reflects differences in layer-wise sensitivity. During memorization localization, the distribution of selected parameters across layers based on small gradients is shown below. We apply the IID condition of CIFAR-10 and conduct a layer sensitivity analysis on ResNet-18. The results indicate that the gradient magnitude naturally reflects differences in layer-wise sensitivity. Moreover, prior research [5] also suggests that shallow layers tend to capture shared patterns or general information, while deeper layers are more likely to learn memorized information. This evidence supports our statement.
>
> | Layer                         | Selected % |
> |-------------------------------|------------|
> | conv1.weight                  | 0.1%       |
> | layer1.0.conv1.weight         | 0.28%      |
> | layer1.0.conv2.weight         | 0.24%      |
> | layer1.1.conv1.weight         | 0.22%      |
> | layer1.1.conv2.weight         | 0.22%      |
> | layer2.0.conv1.weight         | 0.44%      |
> | layer2.0.conv2.weight         | 0.85%      |
> | layer2.0.downsample.0.weight  | 0.05%      |
> | layer2.1.conv1.weight         | 0.83%      |
> | layer2.1.conv2.weight         | 0.85%      |
> | layer3.0.conv1.weight         | 1.68%      |
> | layer3.0.conv2.weight         | 3.35%      |
> | layer3.0.downsample.0.weight  | 0.2%       |
> | layer3.1.conv1.weight         | 3.39%      |
> | layer3.1.conv2.weight         | 3.95%      |
> | layer4.0.conv1.weight         | 11.84%     |
> | layer4.0.conv2.weight         | 24.41%     |
> | layer4.0.downsample.0.weight  | 1.52%      |
> | layer4.1.conv1.weight         | 24.55%     |
> | layer4.1.conv2.weight         | 20.98%     |
> | fc.weight                     | 0.05%      |
> | fc.bias                       | 0.0%       |
>
> **Response to Question 2:**
>
> If the remaining clients and the unlearning clients share a similar data distribution, it implies the absence of non-overlapping information. In this case, although resetting redundant parameters may initially degrade performance relative to the retrained baseline, the subsequent fine-tuning stage can enable the unlearned model to recover its performance. During fine-tuning, the remaining clients can provide equivalent information, allowing the unlearned model to approximate the behavior of the retrained model. Consequently, the similarity in data distributions between the remaining and unlearning clients does not significantly impact the evaluation results.

---

> ### Author Response · Authors · 2025-11-24
> **Feedback for Reviewer hFAH - Part 5**
>
> **Response to Question 3:**
>
> Non-overlapping information and memorization information are both concept definitions at the feature level. Based on relevant studies [3], memorization information refers to unique and atypical data that cannot be generalized to other examples. Because such information cannot facilitate the learning of additional instances, it is difficult to compress and thus must be memorized by neural networks. Conceptually, non-overlapping information represents this unique and atypical content. During training, neural networks are required to memorize non-overlapping information; therefore, non-overlapping information is equivalent to memorization information. Furthermore, for each dataset, there exists unique memorization information and shared information. The shared information is $F_g = F_r \cap F_u$  and the memorization information of $D_u$ is $F_m = F_u - F_g$.
>
> In the context of the federated unlearning problem, where local data or their representations cannot be shared, directly distinguishing between the “overlapping” and “non-overlapping” components in the feature space becomes infeasible. In our approach, rather than explicitly separating these components, we focus exclusively on removing the non-overlapping information from the model parameters. Specifically, we identify and eliminate the redundant parameters associated with $D_r$ that encode non-overlapping information.
>
> Additionally, our GME evaluation results indicate that the top memorization subgroup demonstrates the lowest performance on the retained model. This provides empirical evidence that memorized information corresponds to non-overlapping content, thereby supporting both the internal consistency and practical effectiveness of our framework.
>
> **Response to Question 4:**
>
> Thank you for your question. We answered this question in Response to Weakness 2. We have reasonable theoretical justification and empirical support.
>
> **Response to Question 5:**
>
> If resetting is applied without subsequent fine-tuning, the network's internal representations may become severely disrupted, leading to significant degradation in performance. Therefore, resetting and fine-tuning are inherently inseparable processes. Additionally, it is difficult to determine what information has been forgotten when a neural network fails to work. This is because a very small perturbation to the model's parameters (e.g., in the final fully connected layer) can cause the network to completely fail, not due to actual information loss, but due to a disruption in the cooperation between layers. Fundamentally, in our approach design, the primary forgetting effect is induced by the reset operation, while fine-tuning mainly serves to preserve model fidelity, i.e., to restore the model's generalization ability.
>
> We present the stage results of model performance on CIFAR-10 under both IID and non-IID settings:
> | CIFAR-10 IID | 100–95 | 95–90 | 90–85 | 85–80 | 80–0 | Unlearning Acc. | Test Acc. |
> |--------------------------|--------|-------|-------|-------|------|------------------|-----------|
> | **Only Resetting**       | 7.20 (5.60) | 10.00 (24.40) | 10.80 (66.80) | 8.80 (83.33) | 8.90 (90.22) | 9.52 (80.79) | 9.78 |
> | **After Fine-tuning**    | 20.27 (7.47) | 50.40 (16.00) | 70.93 (6.67) | 85.60 (6.53) | 98.66 (0.46) | 90.56 (0.25) | 74.97 |
> | **Retrained Baseline**   | 12.80 | 34.40 | 77.60 | 92.13 | 99.12 | 90.31 | 74.79 |
>
> |  CIFAR-10 Non-IID | 100–95      | 95–90      | 90–85      | 85–80      | 80–0       | Unlearning Acc. | Test Acc. |
> |----------------------|-------------|------------|------------|------------|------------|------------------|-----------|
> | **Only Resetting**   | 0.93 (11.22) | 0.47 (20.19) | 0.00 (25.55) | 0.00 (42.41) | 0.28 (91.88) | 0.26 (80.15) | 10.00 |
> | **After Fine-tuning**| 23.99 (11.84) | 27.86 (7.20) | 39.72 (14.17) | 47.57 (5.16) | 88.80 (3.36) | 80.16 (0.25) | 74.43 |
> | **Retrained Baseline** | 12.15      | 20.66      | 25.55      | 42.41      | 92.16      | 80.41           | 74.93 |

---

> ### Author Response · Authors · 2025-11-24
> **Feedback for Reviewer hFAH - Part 6**
>
> **Response to Question 6:**
>
> For class-level unlearning, our approach can be easily applied without any modification. Moreover, class-level unlearning is generally easier than client-level unlearning since the overlapping information between different classes is minimal. We conduct class-level unlearning experiments with 10 clients on the CIFAR-10 dataset under both IID and Non-IID conditions to unlearn examples of class 0. The results and details have been demonstrated in **Appendix H.4**. From the results, we observe that our method achieves perfect unlearning performance on class 0 while maintaining strong generalization capability. Specifically, our method achieves unlearning accuracies of 0\% under the two conditions. In terms of generalization performance, the test accuracies of 68.45\% and 71.32\% under IID and Non-IID settings, respectively, outperform the retrained baselines, which achieve 67.52\% and 68.44\% accuracy. Additionally, in terms of time efficiency, our method is 3× faster than the retrained baselines, demonstrating a significant advantage.
>
> | Dataset / Method        | Unlearning Dataset Acc. (%) | Test Dataset Acc. (%) |
> |-------------------------|-----------------------------|------------------------|
> | **CIFAR-10 IID**        |                             |                        |
> | Ours                    | 0.00                        | 68.45                  |
> | Retrained Baseline      | 0.00                        | 67.52                  |
> | **CIFAR-10 Non-IID**    |                             |                        |
> | Ours                    | 0.00                        | 71.32                  |
> | Retrained Baseline      | 0.00                        | 68.44                  |
>
>
> Regarding example-level unlearning, FedMemEraser can be directly applied. For efficiency, the system may accumulate multiple single-example unlearning requests and store them in a request stack. Once the stack reaches capacity or satisfies another predefined condition, the system executes the unlearning algorithm to remove the knowledge of these examples. Following the pipeline of FedMemEraser, we continue to reset redundant parameters with respect to the $D_r$ and fine-tune on the $D_r$. Overall, in our approach, example-level unlearning is essentially similar to client-level unlearning.
>
> | Dataset            | Group (95%,100%]   | Group (90%,95%] | Group (85%,90%]  | Group (80%,85%] | Group (0%,80%]  | Unlearning Dataset Acc. (%) (Δ ↓) | Test Dataset Acc. (%) ↑ | Local Fairness (1e-3) ↓ |
> |--------------------|-----------------------------|-----------------------------|-----------------------------|-----------------------------|------------------------------|-------------------------------------|---------------------------|---------------------------|
> | **CIFAR-10 IID**   |   |   |    |    |   |   |   |    |
> | Ours  | 24.00 ±1.18 (5.33) | 48.67 ±1.68 (5.20) | 65.47 ±4.35 (1.34) | 76.13 ±5.07 (8.14) | 96.44 ±0.90 (2.38)  | 88.37 ±1.08 (1.28)  | 73.83 ±1.04 | 4.06 ±0.76   |
> | Retrained Baseline | 18.67 ±0.75 | 43.47 ±0.50  | 64.13 ±0.19 | 84.27 ±0.19 | 98.82 ±0.07 | 89.65 ±0.06 | 73.84 ±0.30 | 0.11 ±0.03  |
> | **CIFAR-10 Non-IID** |   |   |  |   |   |    |   |   |
> | Ours   | 22.40 ±2.14 (1.00)  | 33.20 ±1.18 (4.00) | 34.27 ±0.83 (2.81) | 39.73 ±2.17 (3.20) | 80.77 ±0.24 (0.29) | 74.20 ±0.23 (0.37) | 73.45 ±0.45 | 54.53 ±8.12  |
> | Retrained Baseline | 21.40 ±0.20   | 29.20 ±0.57  | 31.46 ±0.50  | 42.93 ±1.86 | 81.06 ±0.62  | 74.57 ±0.27  | 74.89 ±0.14              | 25.86 ±15.54|
>
>
>
> **References:**
>
> [1] Adebayo, J., Gilmer, J., Muelly, M., Goodfellow, I., Hardt, M. and Kim, B., 2018. Sanity checks for saliency maps. Advances in neural information processing systems, 31.
>
> [2] Ravikumar, D., Soufleri, E., Hashemi, A. and Roy, K., 2025, October. Towards memorization estimation: Fast, formal and free. In Forty-second International Conference on Machine Learning.
>
> [3] Feldman, V. and Zhang, C., 2020. What neural networks memorize and why: Discovering the long tail via influence estimation. Advances in Neural Information Processing Systems, 33, pp.2881-2891.
>
> [4] Maini, P., Mozer, M.C., Sedghi, H., Lipton, Z.C., Kolter, J.Z. and Zhang, C., 2023, July. Can neural network memorization be localized?. In Proceedings of the 40th International Conference on Machine Learning (pp. 23536-23557).
>
> [5] Yosinski, J., Clune, J., Bengio, Y. and Lipson, H., 2014. How transferable are features in deep neural networks?. Advances in neural information processing systems, 27.
>
> [6] Maennel, H., Alabdulmohsin, I.M., Tolstikhin, I.O., Baldock, R., Bousquet, O., Gelly, S. and Keysers, D., 2020. What do neural networks learn when trained with random labels?. Advances in Neural Information Processing Systems, 33, pp.19693-19704.
>
> [7] Srinivas, S. and Fleuret, F., 2019. Full-gradient representation for neural network visualization. Advances in neural information processing systems, 32.

---

### Official Review · Reviewer_aaJV · 2025-10-30

**Soundness:** 3
**Presentation:** 2
**Contribution:** 3
**Rating:** 6
**Confidence:** 2

**Summary:**

This paper re-examines federated unlearning through the lens of memorization, arguing that common metrics cannot verify whether client-specific information is truly removed. It introduces Grouped Memorization Evaluation (GME): estimate per-sample memorization via multiple retrainings, group the to-be-forgotten samples by score, and contrast “unlearning vs. retraining” within groups for fine-grained assessment. The method FedMemEraser identifies potentially memorization-bearing redundant parameters using the remaining clients’ average gradients, resets them, and fine-tunes only on remaining clients to restore generalization. Experiments on CIFAR-10/100 and EMNIST evaluate GME effectiveness, test performance, local fairness, and time/communication cost.

**Strengths:**

Clear problem reframing and definitions. The decomposition into shared Fg vs. memorized Fm information provides a precise target for Fu and explains why removing overlap harms generalization and fairness. Additionally, the formal “federated memorization unlearning” definition is clean and intuitive.
Evaluation innovation. Grouped Memorization Evaluation (GME) directly probes whether high-memorization examples in Du are “forgotten,” enabling fine-grained, content-aware assessment beyond global accuracy or distance metrics—addressing known shortcomings of prior FU evaluations.
Simple, FL-compatible method. FedMemEraser requires only server-side aggregation of client gradients already present in FL to locate low-update (redundant) parameters, followed by reinit + fine-tuning. This simplicity is attractive and appears robust to both IID and non-IID data.
Broad evaluation coverage. Beyond Grouped Memorization Evaluation and test accuracy, it also reports local fairness and time/convergence behavior.

**Weaknesses:**

Cost and practicality of  Grouped Memorization Evaluation (GME). The memorization score requires training J retrained models without Du to estimate probabilities, which is computationally heavy and may be impractical for large-scale FL. The paper should clarify how J is set.
Hyperparameter sensitivity and selection protocol. The method’s main knob ρ is dataset-specific (e.g., different ρ for CIFAR-10 vs. CIFAR-100), and ablations show strong effects. The paper may be should specify a principled, data-independent selection rule and report sensitivity curves.
Coverage of federated learning in Related Work. The Related Work section barely discusses FL and FU beyond a brief definition in Preliminaries. Please expand the FL-related literature review: position your work against core FL/FU lines (e.g., client heterogeneity, privacy/unlearning, fairness in FL, server-side vs. client-side unlearning), and clarify what is genuinely new here relative to these threads.
Terminology and dataset naming consistency. Please standardize technical terms and dataset names across the paper. For example, consistently use “CIFAR-10/100” (with a hyphen and capitalization) instead of variants like “CIFAR10,” and ensure line 425’s “CIFAR10” is corrected to “CIFAR-10.”

**Questions:**

GME overhead and configuration. What value of JJJ was used to compute memorization scores? How costly is GME relative to a full retrain, and is it strictly an offline evaluation tool (no influence on hyperparameter selection)?

Hyperparameter ρ selection. How should practitioners choose ρ without access to Du? Did you fix ρ per dataset or per run?

Precise definitions in Eqs. (9) and (11). In Eq. (9), what exactly is Pr? Is it the same quantity as M in Eq. (11), or is M derived from Pr (e.g., an average over runs/clients/examples)? Please give a rigorous, self-contained definition , and state the intended direction: in Eq. (11), does lower M indicate better unlearning (i.e., less memorization), or the opposite?
Threshold selection in Eq. (10). How are the thresholds in Eq. (10) chosen in practice (fixed a priori, validated on remaining-client data, or tuned per dataset/model)? Please describe the protocol used in experiments and provide a sensitivity analysis or at least the chosen values and their rationale.
Subgroup mapping and high scores in Table 1. Do the subgroups in Table 1—for example, (95%,100%]—correspond exactly to the thresholds defined in Eq. (10)? If so, why do low-memorization bins show extremely high scores (e.g., EMNIST IID achieves 99.97% in Group (0%,80%])

---

> ### Author Response · Authors · 2025-11-24
> **Feedback for Reviewer aaJV - Part 1**
>
> Thank you for your valuable comments. We will respond to each comment and question in the following.
>
> **Response to Weakness 1 (Cost and practicality of Grouped Memorization Evaluation (GME)):**
>
> Grouped Memorization Evaluation (GME) is an independent fine-grained evaluation framework designed to assess the effectiveness of unlearning. Therefore, GME does not influence our federated unlearning method. In GME, the parameter $J$ serves as a hyperparameter to reduce the variance in memorization scores and mitigate the randomness in the training. Consequently, $J$ is not considered a sensitive hyperparameter and can be selected empirically. In this paper, we set $J = 3$.
>
> The primary computational cost arises from calculating the memorization scores, and this heavy spending has been criticized for a long time.  In this work, we employ memorization scores to adhere to the standard and rigorous memorization evaluation protocol. However, if the computational resources are limited, there exist fast and cost-free memorization proxies that can serve as approximations of the memorization score. The latest proxy is cumulative sample loss (CSL) [1], which tracks how much loss a model accumulates on each sample over the entire training process. Intuitively, easy samples receive low loss early and thus have low CSL, while hard or mislabeled samples accumulate much higher loss. CSL has been shown to closely match formal memorization measures but requires almost no additional computation. We conduct an experiment and demonstrate that CSL could approximate the memorization scores. Specifically, the memorization score and CSL exhibit a 0.82 cosine similarity across the set $D_u$ under the CIFAR-10 IID condition. For detecting the top 20% most memorized examples, the two metrics show an overlap rate of 82%. We have added a new section in the revised version to discuss this solution in **Appendix H.2**.
>
>
>
> **Response to Weakness 2 (Hyperparameter sensitivity and selection protocol):**
>
> The pruning ratio $\rho$ is a key hyperparameter in our method, and we revisit the strategy for selecting $\rho$. The primary factor influencing $\rho$ is the distribution of non-overlapping information within the parameter space. In other words, the proportion of redundant parameters of $D_r$ impacts the selection of $\rho$. Following the fundamental principle that the non-overlapping information of $D_u$ is irrelevant to the important parameters of $D_r$, the main challenge lies in determining an appropriate threshold to identify important parameters of $D_r$. Subsequently, the pruning ratio $\rho$ can be set based on the proportion of redundant parameters.
>
> Here, we propose a method to select $\rho$ based on the distribution of parameter importance. Various indicators can be used to assess parameter importance for a given dataset. For example, the gradient-based input saliency map [2] is a common tool. We apply this tool and analyze the parameter importance distribution on all unlearning tasks and provide the parameter importance distribution figures in **Appendix H.3**. Considering that the magnitudes of gradient updates often span multiple orders of magnitude, we apply the logarithmic function to visualize the distribution.
>
> We measure the parameter importance distributions on three datasets under both IID and Non-IID conditions, and observe similar trends across them. Most parameters exhibit extremely small importance (on the order of $10^{-8}$ to $10^{-6}$), suggesting that they contribute minimally to the current optimization dynamics. As the percentile increases, parameter importance rises gradually and then sharply increases within the top few percent of parameters. This sharp rise typically begins around values of $10^{-5}$. This long-tailed distribution indicates that a small subset of parameters is very important, while the majority remain relatively inactive. Based on this observation, we define parameters with importance values around or above the order of $10^{-5}$ as important. After that, we adaptively select the next lower order of magnitude, i.e., parameters with importance below the order of $10^{-6}$, as redundant parameters. Using this adaptive method, the resulting pruning ratios for CIFAR-10 and EMNIST are approximately 0.4, while for CIFAR-100, the pruning ratio reaches around 0.7. These values are close to our empirical selections.
>
> Additionally, when the task involves unlearning a larger number of clients, the pruning ratio should be increased, as the amount of relevant non-overlapping information grows. Data augmentation also requires a higher pruning ratio, since it promotes feature generalization and tends to enlarge redundant parameters. From the perspective of unlearning effectiveness, it is therefore often preferable to select a larger pruning ratio.

---

> > ### Author Response · Authors · 2025-11-24
> > **Feedback for Reviewer aaJV - Part 2**
> >
> > **Response to Weakness 3 (Weak related work):**
> >
> > Thank you for your comments. We expand the related work and discuss the position of our work.
> >
> > Related work revision:
> > Federated unlearning (FU) extends machine unlearning to federated learning settings. Following a recent survey (Jeong et al., 2024), existing FU methods can be grouped into 4 main categories. The first category is based on historical information, aiming to revert the model to a pre-learning state. A typical approach is SISA (Bourtoule et al., 2021), which clusters clients and removes clusters containing the target clients during the unlearning process. Furthermore, historical information can be used to reconstruct an unlearned model. Following this approach, FedEraser (Liu et al., 2021) enhances model reconstruction efficiency, while FedRecovery (Zhang et al., 2023) improves indistinguishability between unlearned and retained models. Additionally, knowledge distillation can be employed to recover model performance (Wu et al., 2022). The second category involves gradient or weight manipulation, which aims to remove or impair learned representations associated with the data through perturbation (Gu et al., 2024; Fan et al., 2024; Zhao et al., 2023; Deng et al., 2024) or pruning techniques (Wang et al., 2022; Torkzadehmahani et al., 2024). A latest approach (Khalil et al., 2025) proposes NoT, which utilizes weight negation to induce unlearning while preserving model optimality, thereby enabling rapid recovery. The third category focuses on loss function approximation. The unlearning process can be guided using second-order information, such as the Hessian matrix. Liu et al. (2022) proposed leveraging the Fisher information matrix as an approximation of the Hessian to optimize the unlearning process. The fourth category involves reversing the training process. A representative method is gradient ascent (Halimi et al., 2022). Recently, Pan et al. (2025) addressed challenges related to gradient exploration and directional conflicts by proposing FedOSD.
> >
> > Overall, these four technical pathways encompass most existing federated unlearning methods. However, current approaches largely overlook the relationship between unlearning data and remaining data, as well as how this relationship impacts the unlearning process. Additionally, several prior studies (Thudi et al., 2022; Zhang et al., 2024) highlight that current unlearning evaluation protocols are insufficient to reliably verify unlearning effectiveness. Thus, federated unlearning requires a fine-grained evaluation framework to rigorously assess its efficacy.
> >
> >
> > **Response to Weakness 4 (Writing):**
> >
> > We standardize technical terms and dataset names in the revision and carefully check other writing problems.
> >
> > **Response to Question 1:**
> >
> > In this paper, we set $J = 3$. Moreover, GME is an independent federated unlearning evaluation framework and does not influence our unlearning method, FedMemEraser. Therefore, GME should not be used to compare with full retraining, and it serves as an offline evaluation tool.
> >
> > **Response to Question 2:**
> >
> > In this work, we manually choose $\rho$ for specific data distributions. Specifically, we set the pruning ratios to 0.3 and 0.4 for CIFAR-10 under IID and Non-IID conditions, respectively, and apply the same ratios for EMNIST. For CIFAR-100, we adopt pruning ratios of 0.8 and 0.85 for CIFAR-100 under IID and Non-IID settings. Moreover, we give an adaptive method to choose the pruning ratio. The details and results have been discussed in **Appendix H.3**.

---

> ### Author Response · Authors · 2025-11-24
> **Feedback for Reviewer aaJV - Part 3**
>
> **Response to Question 3:**
>
> In Eq. (9), $\Pr(\Phi_G(x_i) = y_i)$ denotes the probability that the example $x_i$ is correctly classified as its ground truth label $y_i$ by the model $\Phi_G$. The term $M$ in Eq. (11) is not related to the probability $\Pr$ in Eq. (9). Moreover, $M$ is used to measure the subgroup or subdataset performance independently. Equations (9) and (10) are solely used to define the subgroups of the unlearning dataset. In contrast, Eq. (11) aims to measure the differences between these subgroups when evaluated on the unlearned model versus the retrained model. In the GME framework, the grouping strategy and the evaluation of each group are decoupled. Thus, for a given subgroup $T_p$, the value $M$ represents the accuracy of the examples within $T_p$. The evaluation for each subgroup is conducted independently.
>
> In Eq. (11), we focus on $\Delta M_{T_p}$, which quantifies the difference in performance between the retrained model and the unlearned model on the subgroup $T_p$. A lower value of $\Delta M_{T_p}$ indicates that the unlearned model behaves more similarly to the retrained model on this subgroup. Therefore, a smaller $\Delta M_{T_p}$ implies more effective unlearning and reduced memorization of the unlearning examples.
>
> For threshold selection in Eq. (10), we use **memorization score percentiles** to group the examples. The thresholds $\{\tau_p\}$ correspond to specific percentiles of the memorization scores. In this work, examples with high memorization scores are approximately associated with non-overlapping information. Therefore, we primarily focus on examples in the top 20\% of memorization scores. Specifically, we divide the top 20\% into four subgroups. Subsequently, we assign the remaining 80\% to a single subgroup because the memorization scores of these examples are nearly 0.  The subgroups presented in Table 1 correspond directly to the thresholds defined in Eq. (10). Of course, the examples could be further partitioned into additional subgroups to enable a more fine-grained evaluation. However, this may not yield additional insights, as our primary goal is to distinguish between non-overlapping and overlapping information in the evaluation. We only need to identify the most memorized examples from general examples. Therefore, grouping thresholds are not sensitive, and the top 20\% can capture enough memorized examples. We give the setting in **Appendix E.2**. Moreover, we provide the average memorization score of each 5% subgroup in CIFAR-10 IID below to support our statement:
>
> Table: Memorization score distribution of $D_u$ in CIFAR-10 IID condition
> | 0–5% | 5–10% | 10–15% | 15–20% | 20–25% | 25–30% | 30–35% | 35–40% | 40–45% | 45–50% |
> |------|--------|---------|---------|---------|---------|---------|---------|---------|---------|
> | 0.0 | 0.0 | 0.0 | 0.0 | 0.0 | 0.0 | 0.0 | 0.0 | 0.0 | 0.0 |
>
> | 50–55% | 55–60% | 60–65% | 65–70% | 70–75% | 75–80% | 80–85% | 85–90% | 90–95% | 95–100% |
> |---------|---------|---------|---------|---------|---------|---------|---------|---------|----------|
> | 0.0 | 0.0 | 0.0 | 0.0 | 0.0 | 0.01 | 0.04 | 0.25 | 0.81 | 0.99 |
>
> Therefore, in practice, the thresholds for grouping are fixed and predefined.
>
> Another relevant question concerns why the low-memorization bins exhibit exceptionally high performance scores. According to [3], examples with low memorization scores tend to be typical examples that carry shared information and generalizable patterns. As noted in [4], such examples are often considered "easy" examples and can be effectively learned by neural networks. Consequently, these low-memorization score examples tend to achieve exceptionally high performance.
>
> **References:**
>
> [1] Ravikumar, D., Soufleri, E., Hashemi, A. and Roy, K., 2025, October. Towards memorization estimation: Fast, formal and free. In Forty-second International Conference on Machine Learning.
>
> [2] Adebayo, J., Gilmer, J., Muelly, M., Goodfellow, I., Hardt, M. and Kim, B., 2018. Sanity checks for saliency maps. Advances in neural information processing systems, 31.
>
> [3] Feldman, V. and Zhang, C., 2020. What neural networks memorize and why: Discovering the long tail via influence estimation. Advances in Neural Information Processing Systems, 33, pp.2881-2891.
>
> [4] Arpit, D., Jastrzębski, S., Ballas, N., Krueger, D., Bengio, E., Kanwal, M.S., Maharaj, T., Fischer, A., Courville, A., Bengio, Y. and Lacoste-Julien, S., 2017, July. A closer look at memorization in deep networks. In International conference on machine learning (pp. 233-242). PMLR.

---

> > ### Comment · Reviewer_aaJV · 2025-11-25
> >
> > Thanks to the author for the detailed reply, I think this is a meaningful work for Federated unlearning community.

---

> > > ### Author Response · Authors · 2025-11-26
> > > **Thank You for Your Follow-Up and Consideration**
> > >
> > > Dear Reviewer aaJV,
> > >
> > > Thank you very much for your follow-up and for confirming that our responses have addressed your concerns. We truly appreciate your time and constructive feedback throughout the process.
> > >
> > > Sincerely,
> > > Authors

---

### Official Review · Reviewer_QEnr · 2025-11-01

**Soundness:** 3
**Presentation:** 3
**Contribution:** 2
**Rating:** 4
**Confidence:** 3

**Summary:**

This paper explores federated unlearning from the perspective of memorization and proposes FedMemEraser, a lightweight method combining gradient-threshold-based pruning and fine-tuning to remove memorized information while retaining shared knowledge. The study provides both theoretical insight into the connection between memorization and unlearning effectiveness and empirical results

**Strengths:**

- Writing is good and easy to follow and understand.

- It provides a clear theoretical formulation linking memorized knowledge to model parameters, which helps explain the trade-off between unlearning effectiveness and performance retention.

**Weaknesses:**

- Why [1] can not become a baseline of this paper? Please demonstrate the experiment results.

- I have carefully checked, and some relevant studies are not discussed in this paper. For example [1] and so on. Please do a comprehensive investigation.

- The proposed FedMemEraser essentially follows a combination of gradient-threshold-based redundant-parameter pruning and a fine-tuning procedure. Compared with existing unlearning approaches based on weight importance or influence functions, its novelty is a problem.

- This paper conducts experiments with a fixed setup of 10 clients and does not evaluate the method under different numbers of clients, leaving its scalability and stability unverified.

[1] Not:Federated unlearning via weight negation

**Questions:**

See in weakness

---

> ### Author Response · Authors · 2025-11-24
> **Feedback for Reviewer QEnr - Part 1**
>
> Thank you for your constructive comments and questions. We will respond to each comment and question in the following.
>
> **Response to Weakness 1:**
> Thank you for your comment. We appreciate you bringing reference [1] to our attention, as it is a concurrent work that had not been included at the time of writing. Accordingly, we have incorporated reference [1] as a baseline in the revised version.
>
> Basically, NoT is a novel federated unlearning approach that aims to apply strong and resilient weight negation to model weights in order to induce unlearning and enable rapid recovery. However, NoT still overlooks the relationship between unlearning data and remaining data that may retain some memorization information of the unlearned data.
>
> Experiment results of [1]:
> | Dataset | Method | Group (95%,100%] Acc (%) (Δ) | Group (90%,95%] Acc (%) (Δ) | Group (85%,90%] Acc (%) (Δ) | Group (80%,85%] Acc (%) (Δ) | Group (0%,80%] Acc (%) (Δ) | Unlearning Acc. (%) (Δ) | Test Acc. (%) | LocalFairness(10^-3)|
> |-------------------|--------------------|------------------------------|-----------------------------|-----------------------------|-----------------------------|----------------------------|--------------------------|---------------|------------------------|
> |CIFAR-100IID| |      |     |     |     |    |  |   ||
> || NoT    | 54.80 ±2.04 (31.60)   | 55.33 ±1.36 (28.26)  | 53.20 ±1.31 (16.80)  | 50.93 ±1.15 (3.33)   | 83.18 ±0.99 (7.96)  | 79.79 ±1.03 (1.58)| 54.72 ±0.49   | 10.85 ±0.94|
> || Ours   | 30.80 ±0.86 (7.60)    | 28.67 ±1.64 (1.60)   | 39.20 ±1.73 (2.80)   | 48.53 ±2.22 (0.93)   | 89.47 ±0.27 (1.67)  | 80.80 ±0.24 (0.57)| 63.33 ±0.31   | 5.20 ±1.36 |
> || Retrained Baseline | 23.20 ±9.62      | 27.07 ±5.85     | 36.40 ±2.90     | 47.60 ±1.18     | 91.14 ±0.68    | 81.37 ±0.51  | 62.29 ±0.58   | 6.46 ±1.81 |
> | CIFAR-100 Non-IID | |      |     |     |     |    |  |   ||
> || NoT    | 46.96 ±2.13 (25.22)   | 40.58 ±0.82 (17.68)  | 45.94 ±0.74 (19.85)  | 40.14 ±1.75 (11.01)  | 64.74 ±0.10 (0.54)  | 63.40 ±0.13 (2.10)| 57.06 ±0.16   | 50.08 ±11.57    |
> || Ours   | 27.25 ±1.25 (5.51)    | 27.39 ±2.22 (4.49)   | 29.71 ±1.68 (3.62)   | 30.43 ±2.48 (1.30)   | 63.44 ±0.69 (1.84)  | 60.37 ±0.74 (0.93)| 62.99 ±0.53   | 42.78 ±6.56|
> || Retrained Baseline | 21.74 ±9.33      | 22.90 ±8.58     | 26.09 ±4.00     | 29.13 ±1.28     | 65.28 ±1.22    | 61.30 ±0.26  | 63.89 ±0.28   | 23.92 ±5.25|
> | CIFAR-10 IID      | |      |     |     |     |    |  |   ||
> || NoT    | 36.40 ±1.70 (23.60)   | 66.27 ±0.19 (31.87)  | 82.00 ±1.31 (4.40)   | 85.20 ±1.42 (6.93)   | 96.22 ±0.87 (2.90)  | 91.06 ±0.75 (0.75)| 74.56 ±0.28   | 6.16 ±0.97 |
> || Ours   | 20.27 ±1.54 (7.47)    | 50.40 ±3.44 (16.00)  | 70.93 ±2.29 (6.67)   | 85.60 ±0.86 (6.53)   | 98.66 ±0.20 (0.46)  | 90.56 ±0.35 (0.25)| 74.97 ±0.27   | 0.91 ±0.07 |
> || Retrained Baseline | 12.80 ±9.34      | 34.40 ±19.80    | 77.60 ±14.71    | 92.13 ±5.58     | 99.12 ±0.62    | 90.31 ±0.16  | 74.79 ±0.38   | 0.09 ±0.01 |
> | CIFAR-10 Non-IID  | |      |     |     |     |    |  |   ||
> || NoT    | 30.22 ±0.44 (18.07)   | 36.62 ±1.01 (15.96)  | 43.15 ±0.58 (17.60)  | 55.87 ±1.01 (13.46)  | 88.15 ±0.67 (4.01)  | 80.74 ±0.45 (0.33)| 72.16 ±0.78   | 76.15 ±12.65    |
> || Ours   | 23.99 ±2.86 (11.84)   | 27.86 ±1.81 (7.20)   | 39.72 ±0.66 (14.17)  | 47.57 ±2.18 (5.16)   | 88.80 ±1.17 (3.36)  | 80.16 ±0.81 (0.25)| 74.43 ±0.78   | 44.12 ±5.20|
> || Retrained Baseline | 12.15 ±8.80      | 20.66 ±14.61    | 25.55 ±18.07    | 42.41 ±10.43    | 92.16 ±4.27    | 80.41 ±0.09  | 74.93 ±0.32   | 30.78 ±6.58|
> | EMNIST IID | |      |     |     |     |    |  |   ||
> || NoT    | 48.50 ±0.14 (1.11)    | 75.33 ±1.06 (22.89)  | 91.11 ±0.08 (8.72)   | 96.67 ±0.00 (3.33)   | 99.66 ±0.00 (0.34)  | 94.45 ±0.05 (2.79)| 92.62 ±0.04   | 4.04 ±0.03 |
> || Ours   | 53.06 ±1.86 (5.67)    | 94.33 ±0.85 (3.89)   | 99.50 ±0.14 (0.33)   | 99.83 ±0.00 (0.17)   | 99.97 ±0.02 (0.03)  | 97.19 ±0.11 (0.05)| 93.50 ±0.06   | 0.70 ±0.15 |
> || Retrained Baseline | 47.39 ±1.95      | 98.22 ±1.23     | 99.83 ±0.14     | 100.00 ±0.00    | 100.00 ±0.00   | 97.24 ±0.01  | 93.60 ±0.01   | 0.18 ±0.03 |
> | EMNIST Non-IID    | |      |     |     |     |    |  |   ||
> || NoT    | 38.88 ±0.25 (18.31)   | 65.23 ±0.17 (1.68)   | 86.07 ±0.70 (1.89)   | 95.77 ±0.15 (1.31)   | 99.80 ±0.01 (0.03)  | 93.37 ±0.04 (0.39)| 89.30 ±0.03   | 22.64 ±3.81|
> || Ours   | 33.33 ±0.76 (12.76)   | 63.50 ±1.43 (3.41)   | 85.45 ±1.22 (2.51)   | 96.59 ±0.06 (0.49)   | 99.80 ±0.01 (0.03)  | 93.16 ±0.24 (0.18)| 89.21 ±0.10   | 19.35 ±1.41|
> || Retrained Baseline | 20.57 ±13.60     | 66.91 ±6.01     | 87.96 ±2.41     | 97.08 ±0.71     | 99.83 ±0.04    | 92.98 ±0.17  | 89.18 ±0.06   | 18.73 ±2.73|

---

> ### Author Response · Authors · 2025-11-24
> **Feedback for Reviewer QEnr - Part 2**
>
> **Response to Weakness 2:**
> Following the reviewer’s comment, we conducted a comprehensive investigation of relevant federated unlearning studies. In the revised manuscript, we provide a detailed subsection in the Related Work section discussing federated unlearning studies.
>
> Related work revision:
> Federated unlearning (FU) extends machine unlearning to federated learning settings. Following a recent survey (Jeong et al., 2024), existing FU methods can be grouped into 4 main categories. The first category is based on historical information, aiming to revert the model to a pre-learning state. A typical approach is SISA (Bourtoule et al., 2021), which clusters clients and removes clusters containing the target clients during the unlearning process. Furthermore, historical information can be used to reconstruct an unlearned model. Following this approach, FedEraser (Liu et al., 2021) enhances model reconstruction efficiency, while FedRecovery (Zhang et al., 2023) improves indistinguishability between unlearned and retained models. Additionally, knowledge distillation can be employed to recover model performance (Wu et al., 2022). The second category involves gradient or weight manipulation, which aims to remove or impair learned representations associated with the data through perturbation (Gu et al., 2024; Fan et al., 2024; Zhao et al., 2023; Deng et al., 2024) or pruning techniques (Wang et al., 2022; Torkzadehmahani et al., 2024). A latest approach (Khalil et al., 2025) proposes NoT, which utilizes weight negation to induce unlearning while preserving model optimality, thereby enabling rapid recovery. The third category focuses on loss function approximation. The unlearning process can be guided using second-order information, such as the Hessian matrix. Liu et al. (2022) proposed leveraging the Fisher information matrix as an approximation of the Hessian to optimize the unlearning process. The fourth category involves reversing the training process. A representative method is gradient ascent (Halimi et al., 2022). Recently, Pan et al. (2025) addressed challenges related to gradient exploration and directional conflicts by proposing FedOSD.
>
> Overall, these four technical pathways encompass most existing federated unlearning methods. However, current approaches largely overlook the relationship between unlearning data and remaining data, as well as how this relationship impacts the unlearning process. Additionally, several prior studies (Thudi et al., 2022; Zhang et al., 2024) highlight that current unlearning evaluation protocols are insufficient to reliably verify unlearning effectiveness. Thus, federated unlearning requires a fine-grained evaluation framework to rigorously assess its efficacy.
>
> **Response to Weakness 3:**
> We understand the reviewer’s concern regarding the novelty of our work. Compared with existing unlearning approaches that rely on weight importance or influence functions, our method offers theoretical novelty and performance advantages.
>
> For theoretical novelty, existing approaches based on weight importance or influence functions primarily focus on eliminating the influence of the unlearning data [1][2]. As a result, they often aim to remove important parameters or high-influence features. However, in our study, we demonstrate that such parameters and features may correspond to general representations or shared patterns, rather than private information.  Removing this important information can degrade the utility of the global model and compromise the contributions of other clients, as discussed in Sections 1 and 4. In contrast to these existing methods, our approach seeks to preserve common knowledge $F_g$ while selectively removing only the truly private information $F_m$ contained in the unlearning dataset. This distinction highlights our theoretical contribution in addressing the critical question of what information should be removed in the context of federated unlearning.
>
> Moreover, our approach also demonstrates superior performance compared to other related techniques, primarily because those methods do not attempt to distinguish between unique and shared information. Among influence-based methods, the approach by Liu et al. [3] is a representative example that utilizes Fisher information to approximate the inverse Hessian matrix for guiding the unlearning process. However, as shown in Table 1, our method outperforms theirs, particularly in the high memorization group. For weight importance-based methods, Table F4 shows that our approach achieves the best performance. Other existing weight importance-based methods tend to prune an excessive number of important neurons, leading to over-unlearning and consequent degradation of model utility. Furthermore, they may inadvertently retain unique information, resulting in ineffective unlearning.

---

> ### Author Response · Authors · 2025-11-24
> **Feedback for Reviewer QEnr - Part 3**
>
> **Response to Weakness 4:**
> We conduct experiments with more clients, and our method continues to perform well in large-scale FL settings and demonstrates non-trivial performance. Specifically, our method achieves performance gaps of 7.47\%, 3.20\%, and 8.00\% on the top memorized subgroup compared to the retrained baselines for configurations with 10, 20, and 50 clients, respectively. Furthermore, the generalization and fairness of the unlearned models remain comparable to those of the retrained baselines across all settings. We provide related details in **Appendix F.5**.
>
> | Client             |  Group (95%,100%]   |   Group (90%,95%]     |  Group (85%,90%]     |  Group (80%,85%]    |   Group (0%,80%]    | Unlearning Dataset Acc. (%) (Δ ↓) | Test Dataset Acc. (%) ↑ | Local Fairness (10⁻³) ↓ |
> |--------------------|--------------------------------------------------------------|-------------------|-------------------|-------------------|-------------------|-------------------------------------|---------------------------|---------------------------|
> | **10 Clients**     |                                                              |                   |                   |                   |                   |                                     |                           |                           |
> | Ours               | 20.27 ±1.54 (7.47)                                       | 50.40 ±3.44 (16.00) | 70.93 ±2.29 (6.67) | 85.60 ±0.86 (6.53) | 98.66 ±0.20 (0.46) | 90.56 ±0.35 (0.25)             | 74.97 ±0.27              | 0.91 ±0.06                |
> | Retrained Baseline | 12.80 ±9.34                                                  | 34.40 ±19.80      | 77.60 ±14.71      | 92.13 ±5.58       | 99.12 ±0.62       | 90.31 ±0.16                        | 74.79 ±0.38              | 0.14 ±0.01                |
> | **20 Clients**     |                                                              |                   |                   |                   |                   |                                     |                           |                           |
> | Ours               | 41.87 ±1.36 (3.20)                                      | 54.13 ±1.51 (6.13)  | 64.80 ±0.65 (5.60) | 71.20 ±1.13 (2.13) | 95.01 ±0.11 (1.56) | 87.56 ±0.23 (1.25)              | 73.56 ±0.18              | 5.22 ±0.08                |
> | Retrained Baseline | 38.67 ±7.64                                                  | 48.00 ±12.26      | 70.40 ±9.13       | 73.33 ±2.38       | 96.57 ±0.04       | 88.81 ±0.30                        | 74.51 ±0.21              | 6.54 ±0.26                |
> | **50 Clients**     |                                                              |                   |                   |                   |                   |                                     |                           |                           |
> | Ours               | 58.00 ±3.27 (8.00)                                       | 60.67 ±1.89 (1.34)  | 62.67 ±0.94 (7.33) | 74.00 ±1.63 (1.33) | 91.00 ±0.28 (0.37) | 85.00 ±0.33 (0.40)              | 75.61 ±0.13              | 7.78 ±0.65                |
> | Retrained Baseline | 50.00 ±8.83                                                  | 59.33 ±10.94      | 70.00 ±4.63       | 72.67 ±3.94       | 90.63 ±0.10       | 84.60 ±0.24                        | 74.88 ±0.11              | 9.34 ±0.19                |
>
>
> References:
>
> [1] Halimi, A., Kadhe, S.R., Rawat, A. and Angel, N.B., 2022, July. Federated Unlearning: How to Efficiently Erase a Client in FL?. In International Conference on Machine Learning.
>
> [2] Pan, Z., Wang, Z., Li, C., Zheng, K., Wang, B., Tang, X. and Zhao, J., 2025, April. Federated unlearning with gradient descent and conflict mitigation. In Proceedings of the AAAI Conference on Artificial Intelligence (Vol. 39, No. 19, pp. 19804-19812).
>
> [3] Liu, Y., Xu, L., Yuan, X., Wang, C. and Li, B., 2022, May. The right to be forgotten in federated learning: An efficient realization with rapid retraining. In IEEE INFOCOM 2022-IEEE conference on computer communications (pp. 1749-1758). IEEE.

---

### Author Response · Authors · 2025-11-25
**Summary of Revisions**

We thank the reviewers for their insightful and constructive feedback. We have revised the manuscript thoroughly in response to all comments. All changes in the revised version appear in blue text. A detailed, point-by-point response is provided in the rebuttal. Below we summarize our main revisions:

1. **Addition of Baselines:** We introduce a new baseline, NoT, a novel method that applies weight negation to model parameters to induce unlearning and enable rapid recovery.
2. **Related Work Revision:** Following the reviewers’ suggestions, we have conducted a comprehensive investigation of related works in federated unlearning and expanded our discussion accordingly.
3. **Explore Large-scale FL Settings:** We extend our evaluation to larger-scale federated learning scenarios with an increased number of clients. The results demonstrate that our method performs well under large-scale FL settings. Detailed results are presented in **Appendix F.5**.
4. **Fast Memorization Proxy:** To address the computational overhead of calculating memorization scores, we can apply a latest proxy metric, Cumulative Sample Loss (CSL), as a fast and efficient alternative. We demonstrate that CSL can effectively replace the memorization score in our setting. Further details are provided in **Appendix H.2**.
5. **Pruning Ratio Selection:** We propose a method based on parameter importance to guide the selection of pruning ratios. Implementation details are included in **Appendix H.3**.
6. **Class-level and Example-level Unlearning:** We examine the applicability of our method to both class-level and example-level unlearning tasks. Our approach proves effective for class-level unlearning. For example-level unlearning, the conditions are similar to client-level unlearning. Results and discussion are available in **Append H.4**.
7. **Additional Details:**
    - We have added further explanations to clarify our memorization evaluation framework.
    - Figure 1 has been revised with fewer data points to improve readability.

We hope that our revisions can address reviewers’ concerns. If the reviewers have additional comments or concerns, we would be glad to further discuss them during the discussion phase.

---

### Author Response · Authors · 2025-11-29
**Summary for AC**

In accordance with ICLR's recommendation for addressing the current situation, we provide a summary of the essential information for AC.

**Summary of our paper**

Revisiting current federated unlearning (FU) studies, we observe that some methods may inadvertently remove shared features while failing to eliminate uniquely memorized information from the unlearning data. This results in incomplete forgetting and client-level unfairness. Therefore, unlearning should focus exclusively on removing non-overlapping or uniquely memorized information associated with the unlearning data.

To address this issue, we propose an offline evaluation framework, **Grouped Memorization Evaluation (GME)**, which enables fine-grained assessment of unlearned models using memorization score-sorted subgroups. Additionally, we introduce a novel unlearning approach, **Federated Memorization Eraser (FedMemEraser)**, designed to selectively remove unique memorization information from the unlearning data.

Experimental results demonstrate the effectiveness of our evaluation framework and show that our unlearning algorithm outperforms existing baselines in terms of removing memorized information.

**Summary of reviewers’ comments**

**The discussion appears unaffected by the information leakage event. After the event, no reviewers responded or revised their scores. Reviewer aaJV provided a response prior to the event, while reviewers QEnr and hFAH did not reply to our comments during the discussion phase.**

We thank the reviewers for their insightful and constructive feedback. Below, we summarize the main concerns and explain how we have addressed each of them:

**Reviewer QEnr**
-   Weakness 1: Lack of comparison with the NoT [1] baseline

We have included the NoT baseline in our experiments. Our results show that this baseline also overlooks memorization information and retains residual information from the unlearned data.

-  Weakness 2: Incomplete related work discussion

We have substantially improved the related work section by expanding the discussion on federated unlearning and including relevant recent studies.

-  Weakness 3: Limited novelty

Our study identifies a key insight into the unlearning problem at the feature level: effective unlearning requires the removal of non-overlapping information associated with the unlearning data. To this end, we focus on eliminating parameters likely to encode such information. This approach is fundamentally different from prior works.

-  Weakness 4: No evaluation in large-scale FL settings

We have added experimental results under large-scale FL settings involving more clients. The results demonstrate that our method performs well in such scenarios. Details have been provided in Appendix F.5.

**Reviewer aaJV**
- Weakness 1: High computational cost of the memorization score.

We refer to a recent proxy for measuring memorization, the *Cumulative Sample Loss (CSL)*[3], which enables assessment without additional computational overhead. Detailed information is provided in Appendix H.2.

- Weakness 2: Lack of a protocol for selecting the pruning ratio.

We propose a pruning ratio selection protocol based on parameter importance. Specifically, we set a threshold based on an order of magnitude to identify and prune unimportant parameters. Further details are available in Appendix H.3.

- Weakness 3: Weak related work

We have significantly revised the related work section by expanding our discussion on federated unlearning and incorporating recent relevant studies.

- Weakness 4: Writing

We standardize technical terms and dataset names in the revision and carefully check other writing problems.

- Question 1: How is $J$ selected in GME? And is GME an offline tool?

In GME, the parameter $J$ is treated as a hyperparameter to reduce the variance of memorization scores. In our experiments, we set $J = 3$. GME is an offline tool.

- Question 2: How is the pruning ratio $\rho$ selected?

We determine $\rho$ based on the proportion of redundant parameters. The detailed selection protocol is provided in Appendix H.3.

- Question 3: Unclear Details

We provide explanations in comments.

---

> ### Author Response · Authors · 2025-11-29
> **Continue...**
>
> **Reviewer hFAH**
>
> - Weakness 1: No selection protocol for the pruning ratio.
>
> We propose a pruning ratio selection protocol based on parameter importance. Specifically, we use an order-of-magnitude threshold to identify and prune unimportant parameters. More details are provided in Appendix H.3.
>
> - Weakness 2: No explanation of how redundant parameters are selected from the remaining dataset $D_r$.
>
> At the information level, all information in $D_r$ is not relevant to the unique memorization content of $D_u$. Therefore, the unique memorization information from $D_u$ should not be associated with the important parameters of $D_r$. Consequently, the memorization information from the unlearning dataset $D_u$ must be embedded in parameters that are redundant with respect to $D_r$.
>
> - Weakness 3: High computational cost of the memorization score.
>
> We address this by adopting a recent proxy for memorization[3], the *Cumulative Sample Loss (CSL)*, which does not require additional computations. Further information is provided in Appendix H.2.
>
> - Weakness 4: Results in Figure 1 are too densely presented.
>
> We have revised Figure 1 in the manuscript and reduced data points to improve the readability.
>
> - Question 1: Why use average gradient magnitude in the approach, and why is no layer-wise sensitivity analysis performed?
>
> Gradient magnitude is a widely used indicator of parameter importance. A small gradient magnitude suggests that variations in the corresponding parameter have limited influence on the loss, implying that these parameters are redundant for the remaining dataset $D_r$.
>
> Additionally, gradient magnitude inherently captures layer-wise sensitivity, and we further elaborate on this in our response.
>
> - Question 2: What happens if the remaining clients and the unlearning clients share a similar data distribution?
>
> In such a case, there is no non-overlapping information between the clients. If resetting leads to performance degradation, the remaining clients can compensate for the lost knowledge during fine-tuning, as they have equivalent information.
>
> - Question 3: Why is memorized information equivalent to non-overlapping information?
>
> According to memorization studies [2], memorized information refers to unique or atypical data that cannot be generalized to other examples. This type of information does not contribute to learning additional instances and is difficult to compress, leading to memorization by the model. Conceptually, non-overlapping information captures this unique and atypical nature. Therefore, we conceptually define memorized information as equivalent to non-overlapping information.
>
> - Question 4: Why choose redundant parameters?
>
> We provide explanations in Weakness 2.
>
> - Question 5: If only positioning and reset phases are performed without fine-tuning, what would happen?
>
> If resetting is applied without subsequent fine-tuning, the network's internal representations may become severely disrupted, leading to significant and average degradation in performance on each memorization subgroup. Therefore, resetting and fine-tuning are inherently inseparable processes. We provide empirical results in comments.
>
> - Question 6: How does the FedMemEraser method apply to example-level or class-level forgetting in federated learning?
>
> Our method is applicable to both example-level and class-level unlearning tasks. We provide detailed explanations and experimental evidence in Appendix H.4.
>
> References:
>
> [1] Khalil, Y.H., Brunswic, L., Lamghari, S., Li, X., Beitollahi, M. and Chen, X., 2025. NoT: Federated Unlearning via Weight Negation. In Proceedings of the Computer Vision and Pattern Recognition Conference (pp. 25759-25769).
>
> [2] Feldman, V. and Zhang, C., 2020. What neural networks memorize and why: Discovering the long tail via influence estimation. Advances in Neural Information Processing Systems, 33, pp.2881-2891.
>
> [3] Ravikumar, D., Soufleri, E., Hashemi, A. and Roy, K., 2025, October. Towards memorization estimation: Fast, formal and free. In Forty-second International Conference on Machine Learning.

---

> ### Author Response · Authors · 2025-11-29
> **Continue...**
>
> **Summary of Revisions**
>
>
> We have revised the manuscript thoroughly in response to all comments. All changes in the revised version appear in blue text. A detailed, point-by-point response is provided in the rebuttal. Below, we summarize our main revisions:
>
> **Addition of Baselines:** We introduce a new baseline, NoT, a novel method that applies weight negation to model parameters to induce unlearning and enable rapid recovery.
>
> **Related Work Revision:** Following the reviewers’ suggestions, we have conducted a comprehensive investigation of related works in federated unlearning and expanded our discussion accordingly.
>
> **Explore Lagre-scale FL Settings:** We extend our evaluation to larger-scale federated learning scenarios with an increased number of clients. The results demonstrate that our method performs well under large-scale FL settings. Detailed results are presented in **Appendix F.5**.
>
> **Fast Memorization Proxy:** To address the computational overhead of calculating memorization scores, we can apply a latest proxy metric, Cumulative Sample Loss (CSL), as a fast and efficient alternative. We demonstrate that CSL can effectively replace the memorization score in our setting. Further details are provided in **Appendix H.2**.
>
> **Pruning Ratio Selection:** We propose a method based on parameter importance to guide the selection of pruning ratios. Implementation details are included in **Appendix H.3**.
>
> **Class-level and Example-level Unlearning:** We examine the applicability of our method to both class-level and example-level unlearning tasks. Our approach proves effective for class-level unlearning; For example-level unlearning, the conditions are similar to client-level unlearning. Results and discussion are available in **Append H.4**.
>
> **Additional Details:**
> - We have added further explanations to clarify our memorization evaluation framework.
> - Figure 1 has been revised with fewer data points to improve readability.
>
>
> We hope that our revisions can address reviewers’ concerns and help AC make a final decision.
>
>
> In conclusion, we sincerely thank the Area Chair for the time, care, and judgment invested in handling this submission, especially given the disruption caused by the recent security incident. We confirm that we have not used, shared, or attempted to exploit any leaked identity information, and we remain fully committed to upholding OpenReview’s Terms of Use and ICLR’s Code of Conduct. We appreciate the effort to preserve a fair review process and are happy to provide any additional clarifications that would be helpful.

---

### Meta-Review · Area_Chair_Rzna · 2026-01-06

**Summary:**

Some of the major concerns include
1. Limited novelty compared to existing work
2. Limited evaluation
3. More comparison with existing FL/FU works is needed
4. High computational overhead

**Reviewer Concerns:**

Some more experiments have been added. Novelty has not sufficiently addressed.

**Reviewer Scores:**

Unchanged

---

### Decision · Program_Chairs · 2026-01-26

Reject